# Glutathione transferase P1 is modified by palmitate

Vanessa Marensi[1¤a¤b], Megan C. Yap[2], Yuhuan Ji[3¤c], Cheng Lin[3], Luc G. Berthiaume[2], Elaine M. Leslie[1]*

1 Department of Physiology and Membrane Protein Disease Research Group, Faculty of Medicine and Dentistry, University of Alberta, Edmonton, AB, Canada, 2 Department of Cell Biology, Faculty of Medicine and Dentistry, University of Alberta, Edmonton, AB, Canada, 3 Center for Biomedical Mass Spectrometry, Department of Biochemistry & Cell Biology, Boston University Chobanian & Avedisian School of Medicine, Boston University, Boston, MA, United States of America

¤a Current address: Chester Medical School, University of Chester, Chester, United Kingdom
¤b Current address: Department of Biochemistry & Systems Biology, Institute of Systems, Molecular and Integrative Biology, University of Liverpool, Liverpool, United Kingdom
¤c Current address: Resolian Bioanalytics Chongqing, Beibei District, Chongqing, China
* eleslie@ualberta.ca

**Data Availability Statement:** All relevant data are within the manuscript and its Supporting Information files. Minimum data sets have been added as S2_File.

## Abstract

Glutathione transferase P1 (GSTP1) is a multi-functional protein that protects cells from electrophiles by catalyzing their conjugation with glutathione, and contributes to the regulation of cell proliferation, apoptosis, and signalling. GSTP1, usually described as a cytosolic enzyme, can localize to other cell compartments and we have reported its strong association with the plasma membrane. In the current study, the hypothesis that GSTP1 is palmitoylated and this modification facilitates its dynamic localization and function was investigated. Palmitoylation is the reversible post-translational addition of a 16-C saturated fatty acid to proteins, most commonly on Cys residues through a thioester bond. GSTP1 in MCF7 cells was modified by palmitate, however, GSTP1 Cys to Ser mutants (individual and Cys-less) retained palmitoylation. Treatment of palmitoylated GSTP1 with 0.1 N NaOH, which cleaves ester bonds, did not remove palmitate. Purified GSTP1 was spontaneously palmitoylated *in vitro* and peptide sequencing revealed that Cys48 and Cys102 undergo *S*-palmitoylation, while Lys103 undergoes the rare *N*-palmitoylation. *N*-palmitoylation occurs via a stable NaOH-resistant amide bond. Analysis of subcellular fractions of MCF7-GSTP1 cells and a modified proximity ligation assay revealed that palmitoylated GSTP1 was present not only in the membrane fraction but also in the cytosol. GSTP1 isolated from *E. coli*, and MCF7 cells (grown under fatty acid free or regular conditions), associated with plasma membrane-enriched fractions and this association was not altered by palmitoyl CoA. Overall, GSTP1 is modified by palmitate, at multiple sites, including at least one non-Cys residue. These modifications could contribute to regulating the diverse functions of GSTP1.

## Introduction

Glutathione transferases (GSTs) are phase II biotransformation enzymes with well defined roles in detoxification [1, 2]. Biotransformation by GSTs occurs through the nucleophilic

**Funding:** Natural Sciences and Engineering Research Council of Canada (NSERC) to EML, [funding reference number RGPIN-2017-06154], https://www.nserc-crsng.gc.ca. Canadian Institutes of Health Research to EML (CIHR grant MOP-272075) https://cihr-irsc.gc.ca/e US National Institutes of Health (NIH grants P41 GM104603, R24 GM134210) to CL. Alberta Cancer Foundation (grant #27369 to LGB).https://albertacancer.ca None of the funders played any role in the study design, data collection and analysis, decision to publish, or preparation of the manuscript.

**Competing interests:** The authors have declared that no competing interests exist.

**Abbreviations:** BCA, bicinchoninic acid; BSA, bovine serum albumin; CDNB, 1-chloro-2,4-dinitrobenzene; 4XCys→Ala or 4XCys→Ser, Cys-less mutant; CoA, Acyl-Coenzyme A; DAPI, 4′,6-diamidino-2-phenylindole; DNP-SG, 2,4-dinitrophenyl-S-glutathione; FAF, fatty acid free; FBS, fetal bovine serum; GSH, Glutathione; GST, Glutathione transferase; HEK293T, SV40-transformed human embryonic kidney 293; HEPES, 4-(2-hydroxyethyl)-1-piperazineethanesulfonic acid; HRP, horse radish peroxidase; LC-MS/MS, liquid chromatography with tandem mass spectrometry; MCF-7, Michigan Cancer Foundation-7; mAb, monoclonal antibody; MOPS, 3-(N-morpholino) propanesulfonic acid; NEM, *N*-ethylmaleimide; NMTs, *N*-myristoyltransferases; pAb, polyclonal antibody; PATs, palmitoyl acyl transferases; PLA, Proximity ligation assay; PVDF, Polyvinylidene difluoride; TBTA, Tris-(benzyltriazolylmethyl) amine; TCEP, Tris-carboxyethylphosphine.

addition of reduced glutathione (GSH, γ-Glu-Cys-Gly) to electrophilic compounds, generally rendering the electrophiles less reactive and more hydrophilic, consequently, preventing cell damage. The human GSTs are classified as mitochondrial, microsomal and cytosolic, where the cytosolic enzymes are sub-classified as alpha (A), pi (P), omega (O), mu (M), sigma (S), theta (T) and zeta (Z) [1, 3]. GSTP1 is a prevalent and widely distributed GST, found in most human cell types and tissues, with the exception of healthy adult hepatocytes [4, 5].

Further to its role in detoxification, GSTP1 has a myriad of other diverse and important cellular functions [6–10]. For example, GSTP1 acts as a chaperone and regulator of multiple signal transduction molecules in the mitogen activated protein kinase pathway (e.g., CRAF, JNK1, and TRAF2) [6, 11, 12]. GSTP1 is also the major player in post-translational *S*-glutathionylation, critical for the regulation of many proteins, especially during oxidative and nitrosative stress [13–15]. GSTP1 binds and sequesters important biological molecules such as dinitrosyl-dithiol-iron complexes as well as non-substrate ligands [16, 17]. Overall, GSTP1 influences many aspects of cell biology, including cell proliferation, death, survival, and redox homeostasis. GSTP1 levels are associated with multiple human pathologies including multiple aggressive cancers, and over-expression is known to result in drug resistance [7].

GSTP1 has a negative net charge and is a soluble protein [18]. Therefore, it is expected that the protein resides in a hydrophilic environment, such as the cytosol [19, 20]. Consistent with this, GSTP1 is found predominantly in the cytosol, and to a lesser extent mitochondria and nuclei [21, 22]. Surprisingly, endogenous GSTP1 was found in plasma membrane-enriched fractions prepared from the human small cell lung cancer cell line H69AR [23] and HEK293T cells [24]. Further to this, GSTP1 stably expressed in MCF7 (Michigan Cancer Foundation 7 human breast adenocarcinoma) cells was found to associate with the plasma membrane-enriched fraction almost as strongly as the integral membrane protein Na$^+$/K$^+$-ATPase [24]. This led us to the hypothesis that GSTP1 was modified by a lipid and the lipidation process promotes GSTP1 association with cellular membranes.

Post-translational modification can be used as a strategy to coordinate dynamic localization, quaternary and tertiary protein structure, and/or activity. Co- and post-translational modification of proteins by lipids (lipidation) increases hydrophobicity, thus the affinity with cellular membranes. Proteins can localize and compartmentalize dynamically within the cell and these processes are especially important for multi-functional proteins, such as GSTP1. Farnesylation, cholesteroylation, glycosylphosphatidylinositol modification, irreversible fatty acylation (myristoylation) and reversible fatty acylation (palmitoylation) are the best studied types of lipid modifications [25–27].

GSTP1 is unlikely to be farnesylated or myristoylated because its amino acid sequence does not contain a CAAX motif (required for farnesylation) or an NH$_2$-terminal glycine residue in position 2 (required for myristoylation) [28]. GSTP1 cannot be cholesteroylated because this modification occurs in the ER lumen [25], and GSTP1 does not have a signal peptide sequence required for ER translocation. Thus, post-translational modification of GSTP1 by palmitate was a logical lipidation process to investigate.

Palmitoylation is the covalent addition of the 16-C saturated fatty acid palmitate to proteins and can be important for membrane association, protein-protein interactions, and/or protein function [29, 30]. Cys residues are by far the most common palmitoylation sites resulting in a thioester bond and referred to as *S*-palmitoylation [31]. *S*-palmitoylation is known to be reversible and thus can dynamically regulate cellular processes it is involved in. Members of a family of 23 palmitoyl acyl transferases (PATs) can catalyze the acylation of Cys residues of substrate proteins with palmitate are present in the mammalian genome [32]. These PATs are localized to membranes of the endoplasmic reticulum (ER), Golgi complex, and plasma membrane with the catalytic domain facing the cytosol [33].

Palmitoylation of non-Cys residues has been reported, but to a much lesser extent. *O*-palmitoylation of Ser residues can be mediated by transmembrane enzymes inserted to the ER membrane [34]. The most studied *O*-palmitoylated protein is the secreted Wnt3a, which is acylated by fatty acids 14–16 carbons in length at Ser209 [35–37]. *N*-palmitoylation of an NH$_2$-terminal Cys residue via amide linkage was reported for Sonic Hedgehog and mediated within the secretory pathway [38]. To our knowledge there are no reports of Thr palmitoylation but, this modification is considered biochemically possible due to the presence of a hydroxyl group on the side chain [39]. *N*-palmitoylation onto an internal Lys is rare, previously reported in a bacterium [40] and tumour necrosis factor alpha (TNF-α) is acylated on a Lys by myristate [41]. Cys palmitoylation of numerous purified mitochondrial proteins has been shown to occur spontaneously in the presence of palmitoyl-CoA [42–44].

The objectives of the current study were to investigate if human GSTP1 is modified by palmitate and determine if palmitoylation influences its dynamic localization and function.

## Material and methods

### Material

Rabbit polyclonal (pAb) anti-GSTP1 (GS72) was from Oxford Biomedical Research (Rochester Hill, MI), rabbit pAb anti-Na$^+$/K$^+$-ATPase (H-300) was from Santa Cruz Biotechnology (Dallas, TX). The AffiniPure donkey anti-rabbit IgG conjugated to Cy$^{TM}$3 was from Jackson ImmunoResearch Laboratories Inc. (West Grove, PA). The Alexa Fluor® 488 conjugated goat anti-rat IgG and Alexa Fluor® 488 conjugated goat anti-rabbit IgG were from Life Technologies (Grand Island, NY). The horseradish peroxidase (HRP) conjugated NeutrAvidin$^{TM}$ goat anti-rat, anti-mouse, and anti-rabbit IgG were from Thermo Scientific (Rockford, IL). Agarose affinity gel beads conjugated to mouse anti-V5 antibody (V5-10), and the mouse monoclonal antibody (mAb) anti-biotin (BN-34) were purchased from Sigma-Aldrich (Oakville, ON). The rabbit pAb anti-V5 was purchased from Rockland (Limerick, PA).

ω-Alkynyl-palmitate and azido-biotin were synthesized as described [45]. Asp-N was purchased from Promega (Madison, WI) and RapiGest SF®-186001861 was purchased from Waters (Milford, MA). EDTA-free Complete$^{TM}$ protease inhibitor cocktail tablets and X-tremeGENE$^{TM}$9 transfection reagent were purchased from Roche Applied Science (Laval, QC). 3-(N-morpholino) propanesulfonic acid (MOPS), bicinchoninic acid (BCA) assay, neutravidin-HRP, immobilized S-hexyl GSH agarose beads and Geneticin®(G418) were from Thermo Scientific (Rockford, IL). 4-(2-hydroxyethyl)-1-piperazineethanesulfonic acid (HEPES), Acyl-Coenzyme A (CoA) synthetase, Tris-(benzyltriazolylmethyl) amine (TBTA), palmitoyl CoA, Tris-carboxyethylphosphine (TCEP), lithium CoA (LiCoA), 1-chloro-2,4-dinitrobenzene (CDNB), magnesium chloride (MgCl$_2$), 1,4₋ dithiothreitol (DTT), Trisma-base, Tris-HCl, reduced L-glutathione (GSH), bovine serum albumin (BSA), CuSO$_4$, ATP, acetonitrile, *N*-ethylmaleimide (NEM), 4′,6-diamidino-2-phenylindole (DAPI), Duolink® *in situ* detection reagent orange, Duolink® *in situ* PLA® probe anti-mouse PLUS affinity purified donkey anti-mouse IgG (H+L), Duolink® *in situ* proximity ligation assay (PLA) ® probe anti-rabbit MINUS affinity purified donkey anti-rabbit IgG (H+L), Duolink® *in situ* wash buffers, were purchased from Sigma-Aldrich (Oakville, ON, Canada). Polyvinylidene difluoride (PVDF) membrane was purchased from Millipore (Bedford, MA, USA). The human GSTP1 expressed in and purified from *E.coli* (GS70) was purchased from Oxford Biomedical Research (Michigan, USA). DAKO mounting medium was from Agilent Technologies (Santa Clara, CA).

## Cell lines

The human breast adenocarcinoma cell line (MCF7) and the SV40-transformed human embryonic kidney 293 (HEK293T) cells were obtained from the American Type Culture Collection (ATCC) (Rockville, MD). MCF7 and HEK293T cells were maintained in Dulbecco's Modified Eagle's Medium (DMEM) (Sigma-Aldrich, Oakville, ON, Canada) supplemented with 10% or 7.5% fetal bovine serum (FBS), respectively and 4 mM L-glutamine. Routine testing for *Mycoplasma sp*. contamination of cell lines was performed using the ATCC Universal Mycoplasma Testing Kit (Manassas, VA). MCF7 cells were used in the majority of experiments because they are known to have very little endogenous GSTP1 due to *GSTP1* silencing by hypermethylation of the CpG dinucleotides at the 5' transcriptional regulatory region [46].

## Generation of V5 tagged GSTP1 expression vectors

The pBluescriptSK(−) GSTP1 construct was a kind gift from Dr. Philip G. Board (Australian National University, Canberra). pcDNA3.1(+) containing the full-length open reading frame of human GSTP1 [pcDNA3.1(+) GSTP1] was generated as previously described [24]. A construct of pcDNA3.1(+) GSTP1 containing a V5 tag at the NH$_2$-terminus [pcDNA3.1(+) V5_GSTP1] was generated by PCR amplification of pcDNA3.1(+) GSTP1 using the forward primer: 5'...AAAAA*GGATCC*ACCATG<u>GGTAAACCTATTCCTAATCCTCTTCTTGGTCTTGAT</u> <u>TCTACA</u>**CCGCCGTATACCGTGG**...3' with a *BamHI* restriction site indicated in italics, the V5 tag sequence is underlined and the GSTP1 sequence is in bold. The reverse primer used was 5'...CGGGCCCTCTAGA*CTCGAG***TCACTGTTTCCCGTTGCCATTGATGG**...3' with a *XhoI* restriction site indicated in italics and GSTP1 sequence indicated in bold.

A construct of pcDNA3.1(+) GSTP1 containing a V5 tag at the COOH-terminus [pcDNA3.1(+)-GSTP1_V5] was generated by PCR amplification of pcDNA3.1(+)-GSTP1 using the forward primer 5'...TACCGAGCTC*GGATCC***GCCATGCCGCCGTATACCGTG**...3' with a *BamHI* restriction site indicated in italics and the GSTP1 sequence in bold. The reverse primer used was'...TTT*CTCGAG*CTA<u>TGTAGAATCAAGACCAAGAAGAGGATTAGGAATAGGT</u> <u>TTACC</u>**CTGTTTCCCGTTGCCATTGATG**...3' with an *XhoI* site indicated in italics, the V5 tag sequence is underlined and the GSTP1 sequence is in bold. The PCR product and empty pcDNA3.1(+) vector were digested with *BamHI* and *XhoI* and then ligated. The correctness of the pcDNA3.1(+)-V5_GSTP1 and pcDNA3.1(+) GSTP1_V5 constructs were confirmed by sequencing (The Centre for Applied Genomics, The Hospital for Sick Children, Toronto, ON).

## Site-directed mutagenesis

GSTP1 mutants were generated using the Agilent QuikChange Lightning site-directed mutagenesis kit (Mississauga, ON). Mutagenesis was performed using the pcDNA3.1 (+)-V5_GSTP1 construct as the PCR template, according to the manufacturer's instructions. Mutagenic primers [Sigma-Aldrich (Oakville, ON) or Integrated DNA Technology (Coralville, IA)] were designed to generate four individual Cys to Ser mutants (V5_GSTP1-C15S, V5_GSTP1-C48S, V5_GSTP1-C102S, V5_GSTP1-C170S) (primer sequences available upon request). Cys-less V5_GSTP1 mutant were also generated; where all four Cys residues within GSTP1 were mutated to Ser (V5_GSTP1-Cys15/Cys48/Cys102/ Cys170Ser, named V5_GSTP1_4XCys→Ser) or Ala (V5_GSTP1_4XCys→Ala). Lastly, a Lys103Arg mutation was added to the V5_GSTP1_4XCys→Ala and it was named V5_GSTP1_4XCys→Ala/Lys103Arg (primer sequences available upon request). The presence of the mutants and the lack of unintentional mutations were confirmed by sequencing the entire GSTP1 coding region (The Center for Applied Genomics, The Hospital for Sick Children, Toronto, ON).

## Generation of stable MCF7 cell lines expressing wild-type GSTP1 tagged with V5 at the $NH_2$ and COOH termini and V5_GSTP1 mutants

Generation of the MCF7-GSTP1 (untagged) and MCF7-vector stable cell lines was described previously [24]. MCF7 cells stably expressing pcDNA3.1(+)-V5_GSTP1 ($NH_2$ V5-tag), -GSTP1_V5 (COOH V5-tag), -V5_GSTP1_4XCys→Ser or -V5_GSTP1_4XCys→Ala/K103R (named MCF7-V5_GSTP1, MCF7-GSTP1_V5, MCF7-V5_GSTP1_4XCys→Ser, or MCF7-V5_GSTP1-4X_Cys→Ala/Lys103Arg, respectively) were generated as described previously [24], with minor modifications. Briefly, MCF7 cells were seeded at $3x10^5$ cells per well in a 6-well plate and transfected with 1 μg DNA using 3 μl X-tremeGENE$^{TM}$ 9, according to the manufacturer's instructions. Forty-eight hours post-transfection, cells were selected in 1 mg/ml of G418, and approximately two weeks later the selected cells were cloned by limiting dilution. GSTP1 protein levels in G418-resistant cell clones were determined by immunoblotting with the rabbit anti-GSTP1 pAb (1:5000). Cells were then maintained in DMEM with 10% FBS supplemented with 600 μg/ml of G418.

## Transient transfection of MCF7 and HEK293T cells

HEK293T and MCF7 cells were seeded at a density of 6.5 $x10^5$ and $1x10^6$ cells per T-25 flask, respectively. Twenty-four h later, cells were transfected at a 3:1 ratio of X-tremeGENE9:DNA, according to the manufacturer's instructions (Roche Applied Sciences, Penzberg, Germany).

## Metabolic labelling of cells with ω-alkynyl palmitate

Metabolic labelling of cells with ω-alkynyl-palmitate, was adapted from a method described previously [45]. Briefly, HEK293T and MCF7 cells transiently or stably expressing GSTP1 were grown to 70–80% confluency, then the culture medium was exchanged for DMEM with 10% fatty acid free (FAF)-FBS and ω-alkynyl-palmitate (100 μM). The ω-alkynyl-palmitate was dissolved in DMSO and diluted in the FAF medium right before labelling started. The final DMSO concentration in the culture media was 0.1%. The unlabelled cells were maintained in DMEM containing the regular FBS and the same amount of DMSO (vehicle) as the labelled cells. The FAF medium containing ω-alkynyl-palmitate was replaced daily for 24–48 h, unless indicated otherwise.

## Immunoprecipitation of wild-type and mutant forms of GSTP1

After metabolic labelling with ω-alkynyl-palmitate, cells were washed with cold PBS, harvested, and lysed with cold EDTA-free RIPA buffer [0.1% SDS, 50 mM HEPES, pH 7.4, 150 mM NaCl, 1% Igepal CA-630, 0.5% sodium-deoxycholate, 2 mM $MgCl_2$, EDTA-free Complete$^{TM}$ protease inhibitor cocktail] for 15 min at 4˚C. Cell lysates were centrifuged at 16,000 $g$ for 10 min at 4˚C and the post-nuclear supernatant collected. GSTP1 was then immunoprecipitated from this fraction by incubating with agarose beads conjugated to mouse anti-V5 antibody (V5-10) overnight at 4˚C. Beads were then spun down at 16 000 $g$ for 10 min and washed with 1 ml of cold PBS with 0.1% Triton-X, 3–5 times. GSTP1 was eluted from the beads by incubating for >10 min at room temperature with Laemlli buffer or subjected to click chemistry (described below).

## Detection of GSTP1 metabolically labelled with ω-alkynyl-palmitate using click chemistry

Immunoprecipitated GSTP1 was eluted from the agarose beads conjugated to mouse anti-V5 antibody (V5-10), by incubating at 80˚C for 15 min in 50 mM HEPES, pH 7.4, with 1% SDS.

The beads were then pelleted by centrifuging at 2800 g and the supernatant containing GSTP1 collected. The supernatant was then incubated with click reaction components [Tris-(benzyl-triazolylmethyl) amine (100 μM), $CuSO_4$ (1 mM), Tris-carboxyethylphosphine (1 mM), and azido-biotin (100 μM)] at 37°C for 30 min, protected from light. The click reaction was stopped by the addition of a 10X volume of ice-cold acetone and incubated overnight at -20°C to precipitate proteins. To determine the alkali sensitivity of palmitoylation, samples were treated with NaOH (0.1 M) at room temperature for 1 h after being eluted from the beads. The samples were then neutralized by adding an equal volume of HCl (0.1 M). Proteins were then processed and subjected to click chemistry as described above. Precipitated proteins were then pelleted at 16 000 g for 15 min, the supernatant discarded and the pellet resuspended in Laemmli buffer containing DTT (20 mM) (Yap et al., 2009). Samples were heated at 95°C for 5 min, resolved on 11% SDS-PAGE then electrotransfered to a PVDF membrane. Biotinylated ω-alkynyl-palmitate-labelled GSTP1 was detected with neutravidin-HRP (1:20000) and total GSTP1 loading was detected with rabbit pAb anti-GSTP1 (GS72) (1:5000).

## Identification of GSTP1 proximity with ω-alkynyl-palmitate using in situ proximity ligation assay (PLA)

MCF7-V5_GSTP1and MCF7-vector cells were seeded at $2x10^5$ cells per well of a 6-well plate on glass cover slips and grown overnight. Cells were then metabolically labelled with ω-alky-nyl-palmitate for 24–48 h. Coverslips were washed twice with PBS, fixed with ice-cold metha-nol at -20°C, and permeabilized with Triton-X 100 (0.1%) in PBS for 15 min at room temperature. Cells were washed twice with 1X PBS and then the click reaction components were added onto the coverslips and incubated at 37°C for 1 h in the dark. After the click reaction, coverslips were washed three times with 1X PBS and incubated with blocking buffer [BSA (3%) + Triton-X 100 (0.1%) in 1X PBS] for 1 h at room temperature. Rabbit pAb anti-GSTP1 (GS72) (1:250) and mouse mAb anti-biotin (BN-34) (1:500) were diluted in blocking buffer and added to the coverslips for 1–2 h at room temperature. Cells were washed with washing buffer [BSA (0.3%) + Triton (0.1%) in 1X PBS] and from this step on, reagents from the DUOLINK® PLA kit were used according to the manufacturer's instructions (OLINK Bio-science, Uppsala, Sweden). Briefly, coverslips were washed with washing buffer A, probed with anti-mouse PLUS and anti-rabbit MINUS 1:500, where PLUS and MINUS stands for 5' to 3' reverse and forward oligonucleotides, respectively, and incubated for one hour at 37°C. Cover-slips were washed with washing buffer A and incubated with 1 U of ligase in ligation buffer for 30 min at 37°C, then washed twice with washing buffer A. The ligation product was amplified with 5 U of polymerase in amplification buffer containing the orange detection reagent [exci-tation 554 nm and emission at 579 nm ($Cy^{TM}3$)] and DAPI, for 100 min at 37°C. All of the reactions with the PLA kit were done in a humidified chamber. Total GSTP1 was detected by the addition of Alexa Fluor® 488 conjugated goat anti-rabbit IgG (1:500) during the PLA amplification step. Coverslips were mounted onto slides with DAKO mounting medium and stored at 4°C protected from light until visualization. MCF7-GSTP1 cells subjected to the PLA, as described above, but without ω-alkynyl-palmitate metabolic labeling or without being sub-jected to click chemistry were included as negative controls, in addition to the MCF7-vector cell line.

## Immunofluorescence imaging

Cells were viewed with a 60X oil immersion objective with a fully motorized inverted fluores-cence microscope (Leica DMI6000 B, Wetzlar, Germany) coupled to a X-Cite® exacte fluores-cence light source, equipped with Quorum Mac 6000 system and the Angstrom illumination

system associated with Optigrid (Guelph, ON, Canada). The MetaMorph software (Molecular devices, Sunnyvale, CA) was used for multi-dimensional acquisition that allows the measurement of multiple Z sections. Images were captured with a Flash 4.0 camera (Hamamatsu) and analyzed by Volocity® (Improvision, Perkin Elmer, Waltham, MA).

## Synthesis of ω-alkynyl-palmitate-CoA

ω-alkynyl-palmitate-CoA was freshly synthesized prior to use in experiments, as described previously [47]. Briefly, ω-alkynyl-palmitate (1 mM) was incubated with Li CoA (1 mM), ATP (5 mM), Triton X-100 (2.5%), and acyl CoA synthetase (1 U/ml) in acylation buffer (Tris-HCl 20 mM, pH 7.4, $MgCl_2$ 10 mM, EGTA 200 μM, DTT 2 mM) at 37˚C for 30 min protected from light.

## Characterization of PAT-independent modification of GSTP1 with ω-alkynyl palmitate-CoA and detection using click chemistry

Commercially available recombinant human GSTP1 (0.5 μg) expressed in and purified from *E. coli* was incubated with ω-alkynyl-palmitate-CoA (100 μM) for 20 min at room temperature for *in vitro* palmitoylation. After the incubation, click reaction components [Tris-(benzyltriazolylmethyl) amine (100 μM), $CuSO_4$ (1 mM), Tris-carboxyethylphosphine (1 mM), and azido-biotin (100 μM)] in HEPES (50 mM) containing SDS (1%) were added and incubated at 37˚C for 30 min, protected from light. The click reaction was stopped and GSTP1 precipitated by the addition of a 10X volume of ice-cold acetone overnight at -20˚C. Precipitated protein was pelleted at 16 000 *g* for 15 min at 4˚C and resuspended in Laemmli buffer containing DTT (20 mM). Samples were heated at 95˚C for 5 min then resolved on 11% SDS-PAGE and electrotransfered to a PVDF membrane.

As controls, to detect possible non-specific non-covalent interactions of ω-alkynyl-palmitate-CoA with GSTP1, GSTP1 was incubated with either ω-alkynyl-palmitate or LiCoA alone, as described above for ω-alkynyl-palmitate-CoA.

## Biochemical characterization of PAT-independent GSTP1 palmitoylation

To determine if the PAT-independent covalent modification of GSTP1 by ω-alkynyl-palmitate is through an oxyester, thioester, or amide bond, biochemical aproaches previously applied to characterize palmitoylation of other proteins were utilized [47]. To begin with, the functional group on potentially palmitoylated amino acids Cys and Lys were blocked with the covalent modifier NEM at either pH 7.4 (to block Cys residues) or pH 10 (to block Lys and Cys residues). Thus, GSTP1 from *E. coli* (1 μg) was incubated in the dark with or without NEM (10 mM) in MOPS (50 mM at either pH 7.4 or 10), at room temperature, for 30 min. NEM-treated GSTP1 was then reacted with ω-alkynyl-palmitoyl-CoA, ω-alkynyl-palmitate, or Li-CoA, and subjected to the click reaction. To determine if GSTP1 was covalently modified by palmitate through an ester bond, additional treatments were done. NaOH breaks oxyester and thioester bonds, but not amide bonds. Palmitoylated GSTP1 (2 μg) was incubated with NaOH (0.1 M) for one h at room temperature. NaOH treatment was stopped by the addition of HCl (at a 1:1 molar ratio with NaOH). GSTP1 was precipitated with ice-cold acetone overnight and subjected to click chemistry followed by 11% SDS-PAGE and immunoblotting, as described above.

## Mapping non-PAT mediated GSTP1 palmitoylation sites by LC-MS/MS

GSTP1 (2 μg) was incubated with palmitoyl-CoA (100 μM) or CoA (100 μM) for 20 min at room temperature, and then digested for 6 h with 0.6 μg Asp-N (which cuts on the $NH_2$-

terminal side of Asp and Cys residues) in 0.05% surfactant RapiGest SF® at 37°C. The digestion was stopped with trifluoroacetic acid (TFA, 1%) and dried using a speed vacuum. The samples were enriched for palmitoylated peptides using column chromatography with POROS R1 50 resin. Palmitoylated peptides were eluted stepwise in 60% and 80% acetonitrile and analyzed by liquid chromatography, using a C18 analytic column and analyzed by electrospray tandem mass spectroscopy (LC/MS/MS) on a LTQ-Orbitrap XL mass spectrometer as described previously (Ji *et al.*, 2013). CID tandem mass spectra of peptides of interests were processed by the XCalibur software (Thermo Fisher Sceintific), and manually interpreted with assistance from the online proteomics tool ProteinProspector (https://prospector.ucsf.edu/prospector/mshome.htm).

## Subcellular fractionation of MCF7-GSTP1 cells

MCF7-GSTP1 cells ($2 \times 10^7$) were re-suspended in tris-sucrose-buffer (50 mM Tris, pH 7.4, 250 mM sucrose) containing $CaCl_2$ (250 μM) and EDTA free Complete™ protease inhibitors, and disrupted by $N_2$ cavitation as described previously [48]. Cell homogenates were centrifuged at 2000 *g* at 4°C for 10 min to remove nuclei and unbroken cells. Supernatant (500 μl) was transferred to a 1.7 ml ultracentrifuge tubes, centrifuged at 100 000 *g* for 30 min at 4°C (Beckman TLA 110 rotor). Supernatant was collected (490 μl of cytosolic fraction) while cellular membranes (P100) were washed once with 1X PBS and re-suspended in 50 μl Tris (10 mM, pH 7.4). For immunoprecipitation of GSTP1, the cellular membranes were re-suspended in 500 μl Tris (10 mM, pH 7.4) + SDS (0.1%). BCA or Bradford protein assays were performed to determine protein concentrations, and the fractions were either used fresh or stored in aliquots at -80°C (cellular membrane fractions) or -20°C (cytosolic fractions).

## Catalytic activity of total, cytosolic, and crude membrane fractions isolated from MCF7-vector and MCF7-GSTP1

CDNB is a well established and commonly used GST substrate [3] that is conjugated with GSH to form 2,4-dinitrophenyl-S-glutathione (DNP-SG). The enzymatic activity of subcellular fractions (total, cytosol, or cellular membranes) isolated from MCF7-GSTP1 and MCF7-vector cells was evaluated using the CDNB assay. Assays were performed in 96 well plates and activity measured by monitoring the formation of DNP-GS at 340 nm ($\Delta\varepsilon = 9600$ M$^{-1}$ cm$^{-1}$) every 30 seconds for 10 min, in 100 mM potassium phosphate buffer ($K_2PO4$) pH 6.8 at 25°C [3]. CDNB (1 mM) was added to 3 μg of protein (input, cytosol, and membrane), followed by GSH (2.5 mM). Spontaneous reaction of GSH with CDNB (blank) was subtracted from the absorbance of the GSTP1-catalyzed reactions. The rate was calculated over 300 s and expressed as mean ± SD of at least three independent experiments. In parallel with the CDNB, 5 μg of each fraction was analyzed by western blotting and the relative level of GSTP1 in each fraction determined by densitometry using ImageJ software. Equal protein loading was confirmed with amido black, a total protein stain. The final absorbance (mean of absorbance triplicates–mean of blank absorbance triplicate) of CDNB catalytic activity was normalized to GSTP1 level.

## GSTP1 purification from MCF7-GSTP1 cells

The cytosolic fraction of MCF7-GSTP1 was incubated with 300 μl of immobilized *S*-hexyl-GSH-agarose beads for 1 h at 4°C, centrifuged at 10000 *g* for 10 min at 4°C and then beads were washed with $Na_2HPO_4$, pH 6.8, and NaCl 150 mM. Protein was eluted with 20 mM GSH in $Na_2HPO_4$, pH 8.0 and NaCl 150 mM. Eluted protein was concentrated using a speed vacuum, then the protein concentrations were determined at 280 nm using a NanoDrop 2000 (Wilmington, DE, USA). Eluted and concentrated protein (3 μg) was resolved on 11%

SDS-PAGE. The gel was fixed for 30 min in fixation buffer (isopropanol 25%, acetic acid 10%), stained for 2 h with Coomassie 250 (60 mg/L) for 30 min followed by overnight destaining in acetic acid 10%, then the gel was dried.

### Characterization of GSTP1 association with the MCF7 plasma membrane-enriched fraction

Plasma membrane-enriched fractions were prepared from MCF7 cells using $N_2$ cavitation, as previously described [48]. Total protein concentrations were determined using the BCA assay according to the manufacturer's instructions. Human GSTP1 purified from *E.coli* or MCF7-GSTP1 cell cytosol (2 μg) was incubated with or without palmitoyl-CoA (100 μM) in sodium phosphate buffer $K_2HPO_4$ (100 mM, pH 7.4) at 25˚C for 20 min. Plasma membrane-enriched fractions (20 μg of protein) were then added and incubated for 30 min at 37˚C, samples were then centrifuged at 100000 *g* at 4˚C for 1 h. Supernatants were collected and diluted in Laemmli buffer whereas the pellet was washed once with 100 μl of Tris buffer (10 mM, pH 7.4), and then resuspended in Laemmli buffer. Samples were resolved on 11% SDS-PAGE, transferred onto a PVDF membrane and immunostained with rabbit pAb anti-GSTP1 (GS72) (1:5000), equalness of plasma membrane fraction loading was detected using the rabbit pAb anti-$Na^+$/$K^+$-ATPase (H-300) (1:10000).

## Results

### GSTP1 is modified by ω-alkynyl-palmitate in MCF7 cells

To determine if GSTP1 is modified by palmitate, MCF7 cells (which do not express endogenous GSTP1) stably expressing GSTP1 tagged at the $NH_2$-terminus with the V5 epitope (MCF7_V5_GSTP1) were metabolically labelled with ω-alkynyl-palmitate and harvested at multiple time points over four to forty-eight hours. Cells were lysed, GSTP1 was immunoprecipitated with V5 antibody conjugated to agarose beads, eluted from the beads, and subjected to click chemistry to label ω-alkynyl-palmitate with biotin. Samples were analyzed by western blotting, palmitoylated GSTP1 was detected with neutravidin-HRP and total GSTP1 was detected with rabbit pAb anti-GSTP1. We report the novel observation that GSTP1 is modified by the palmitate analogue ω-alkynyl-palmitate (Fig 1A, top panel) as evidenced by the detection of a neutravidin-HRP signal (revealing the presence of a GSTP1-palmitate-biotin complex). This signal was only detected for GSTP1 immunoprecipitated from cells metabolically labelled with ω-alkynyl-palmitate, and not cells treated with a vehicle control. GSTP1 palmitoylation was not detected until eight hours after initiating metabolic labelling with ω-alkynyl-palmitate and plateaued at 24 h (Fig 1A and 1B). Metabollic labelling with ω-alkynyl-palmitate was done for a minimum of 24 h for all future experiments.

Previous studies have shown that a green fluorescence protein tag on the COOH terminus of GSTP1 prevents its targeting to the nucleus, possibly through a conformational change [49]. Thus it was important to test if the V5 tag position might influence palmitoylation. MCF7 cells transfected with GSTP1 tagged with the V5 epitope at either the $NH_2$-terminus (MCF7-V5_GSTP1) or COOH-terminus (MCF7-GSTP1_V5) were metabolically labelled with ω-alkynyl-palmitate, subjected to click chemistry, and analyzed by western blotting. Consistent with the time course data (Fig 1A and 1B), a strong neutravidin-HRP signal was detected for GSTP1 from MCF7-V5_GSTP1 cells metabolically labelled with ω-alkynyl-palmitate (Fig 1C, left most lane). In contrast, no neutravidin signal was detected for GSTP1 from MCF7-GSTP1_V5 cells metabolically labelled with ω-alkynyl-palmitate (Fig 1C third lane) or for either cell line treated with vehicle control (Fig 1C). Thus, the V5 epitope tag at the

A)

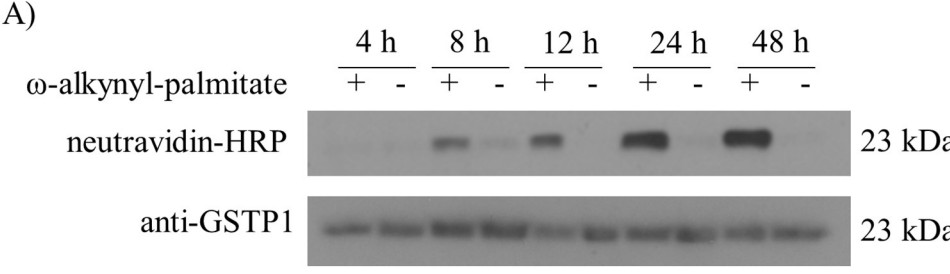

B)

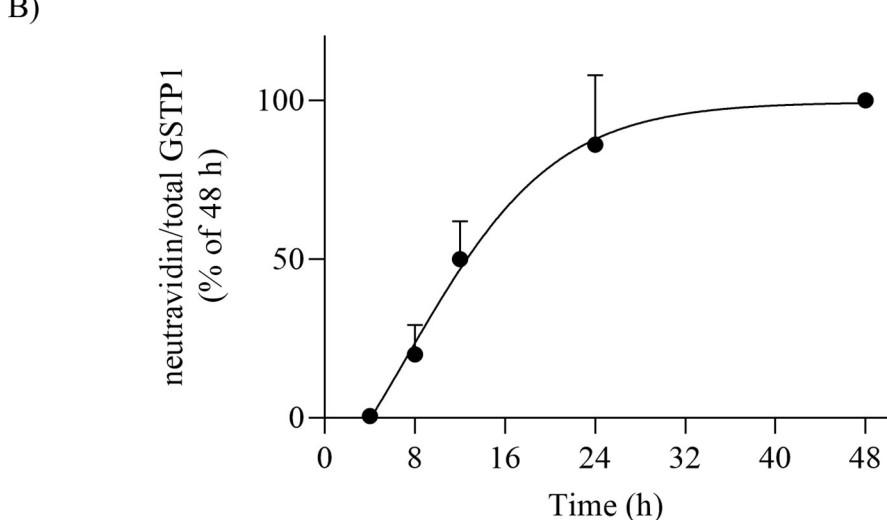

C)

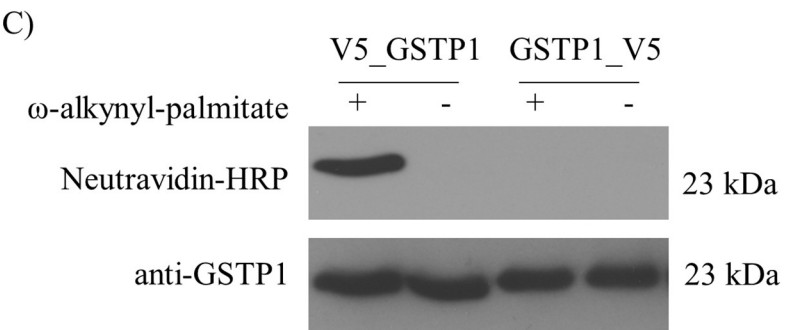

**Fig 1. Palmitoylation of GSTP1 in MCF7 cells.** MCF7 cells stably expressing GSTP1 with a V5 epitope tag at the amino- and carboxy-termini (MCF7-V5_GSTP1 or MCF7-GSTP1_V5, respectively) were metabolically labelled with ω-alkynyl-palmitate (+) or DMSO (-). Cells were lysed and GSTP1 was immunoprecipiated with agarose beads conjugated to mouse anti-V5 antibody (V5-10), subjected to click chemistry, resolved on 11% SDS-PAGE and transferred to a PVDF membrane. Palmitoylated GSTP1 was detected with neutravidin-HRP (1:20000) and total GSTP1 was detected with rabbit pAb anti-GSTP1 (GS70) (1:5000). **(A)** Time course of V5_GSTP1 metabolically labelled for 4, 8, 12, 24 and 48 h. *(Top panel)* Signal from neutravidin-HRP represents palmitoylated V5_GSTP1. *(Bottom panel)* Signal from anti-GSTP1 antibody represents total V5_GSTP1. **(B)** Densitometry on the bands detected by neutravidin-HRP and anti-GSTP1 in (A) was performed using ImageJ software. The level of palmitoylated V5_GSTP1 (neutravidin-HRP) relative to total V5_GSTP1 was expressed as a % of the 48 h time point. Symbols represent means +SD from 3 independent experiments. **(C)** MCF7- V5_GSTP1 and -GSTP1_V5 cells were metabolically labelled with ω-alkynyl-palmitate for 24 h *(Top panel)*. Signal from neutravidin-HRP represents palmitoylated GSTP1. *(Bottom panel)* Signal from anti-GSTP1 antibody represents total GSTP1. Shown is a representative blot and similar results were obtained in three additional independent experiments.

COOH-terminus prevents detectable GSTP1 palmitoylation and the $NH_2$-terminal V5_GSTP1 was used for all future experiments.

## GSTP1 is found in close proximity with ω-alkynyl-palmitate by a modified proximity ligation assay

GSTP1 palmitoylation was further characterized by a specialized adaptation of an *in situ* proximity ligation assay (PLA) for the detection of palmitoylated proteins [36]. MCF7-V5_GSTP1 and MCF7-vector cell lines were metabolically labelled with ω-alkynyl-palmitate, fixed, subjected to click chemistry, and the PLA performed utilizing antibodies directed towards the V5 tag of GSTP1 and biotin (to detect palmitoylated protein). As expected, GSTP1 (green) was distributed throughout the cell and nuclei (Fig 2A.i). The PLA product, indicating a close proximity between GSTP1 and ω-alkynyl-palmitate (orange) was also distributed throughout the cell and nuclei (Fig 2A.ii). Localization of GSTP1 (green) and the PLA product (orange) was overlapping in many regions of the cell including stippled structures around and within nuclei (Fig 2A.iii). Several regions of only GSTP1 (green) were also present, showing the presence of GSTP1 not in close proximity with ω-alkynyl-palmitate. Additional control experiments of MCF7-V5_GSTP1 are shown in Fig 2B. These include images captured from the same experiment (with no Alexa Fluor® 488 secondary to detect total GSTP1) but in the absence of ω-alkynyl-palmitate (Fig 2B.i) and the click chemistry step (Fig 2B.ii). Results were comparable with MCF7-vector cells (Fig 2A (iv to vi) compared with 2B), as expected for a specific PLA reaction. Overall, these data provide evidence that specific populations of GSTP1 are palmitoylated.

## Individual GSTP1 Cys to Ser mutants retain palmitoylation

*S*-palmitoylation of Cys residues is the most common palmitoylation type. GSTP1 has four Cys residues (Cys15, Cys48, Cys102, and Cys170) the location of each is indicated in the cytosolic GSTP1-1 dimer X-ray crystal structure (bound to the GSH conjugate of ethacrynic acid) (Fig 3A). To determine if any of the Cys residues are modified by palmitate they were individually mutated to Ser (V5_GSTP1-Cys15Ser, V5_GSTP1-Cys48Ser, V5_GSTP1-Cys102Ser, and V5_GSTP1-Cys170Ser) and expressed transiently in HEK293T cells. Transfected cells were metabolically labelled with ω-alkynyl-palmitate, GSTP1 was immunoprecipitated using agarose beads conjugated to mouse anti-V5 antibody, then subjected to click chemistry and analyzed by western blotting with neutravidin-HRP and rabbit pAb anti-GSTP1 (Fig 3B). If GSTP1 contains a single Cys residue that is palmitoylated it was expected that a complete loss in neutravidin signal would be observed for one of the Cys to Ser mutants. Somewhat surprisingly, all four individual Cys to Ser mutants retained a neutravidin/ palmitoylation signal. Each mutant had its specific negative control (cells treated with vehicle in place of ω-alkynyl-palmitate), and as expected, no signal was detected.

## Cys-less GSTP1 retains palmitoylation and GSTP1 palmitoylation is resistant to NaOH

To investigate the possiblity that multiple Cys residues in GSTP1 are palmitoylated, a Cys-less GSTP1 mutant was generated by mutating all Cys residues to Ser [Cys15/45/102/170Ser-GSTP1 (V5_GSTP1_4XCys→Ser)]. MCF7 cells stably expressing this construct [MCF7-V5_GSTP1_4XCys→Ser] were metabolically labelled with ω-alkynyl-palmitate, immunoprecipitated with agarose beads conjugated to mouse anti-V5 antibody, subjected to click chemistry, and analyzed by western blotting with neutravidin-HRP and rabbit pAb anti-

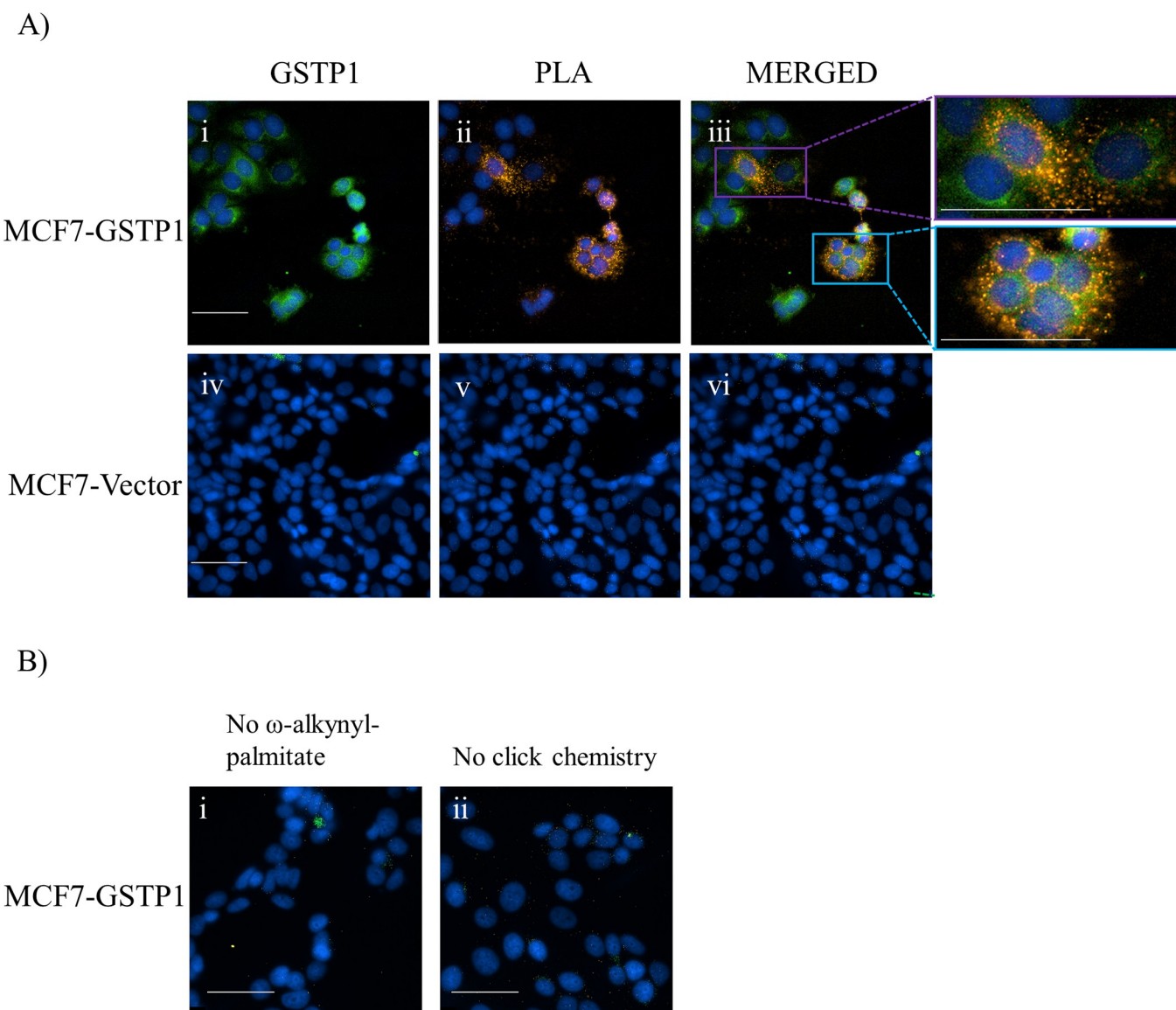

**Fig 2. Detection of cellular GSTP1 palmitoylation by a modified proximity ligation assay.** MCF7-V5_GSTP1 and MCF7-vector cells were metabolically labelled with ω-alkynyl-palmitate for 24 h, fixed, permeabilized and subjected to click chemistry. Cells were incubated with mouse mAb anti-biotin (BN-34) (1:500) and pAb rabbit anti-GSTP1(GS72) (1:250) for reaction with the biotinylated fatty acid analogue and GSTP1, respectively. Cells were then incubated with secondary antibodies PLUS and MINUS probe and ligated. The ligation product was amplified and detected with Duolink® *in situ* detection reagent (orange), nuclei (blue) were detected with DAPI and GSTP1 (green) was detected with anti-rabbit Alexa488 (1:500). Cells were viewed with a Leica fluorescence microscope at a 60X magnification. **(i-iii)** MCF7-V5_GSTP1 cells **(iv-vi)** MCF7-vector control (negative control). (B) Images captured from the same experiment as A (with no Alexa Fluor® 488 secondary to detect total GSTP1) (i) in the absence of ω-alkynyl-palmitate and (ii) with no click chemistry step. Images were acquired in Volocity. White bars indicate 20 μm. Similar results were obtained for multiple cells from two independent experiments.

GSTP1. The Cys-less mutant retained a strong palmitoylation signal (Fig 3C). Ester bonds are susceptible to alkaline hydrolysis and treatment with NaOH is a useful tool for characterizing the nature of the chemical bond of protein acylation [47]. To determine if V5_GSTP1-WT and V5_GSTP1-4XCys→Ser are modified by palmitate through an ester bond (either oxy- or thioester), samples were treated with NaOH. Palmitoylation of V5_GSTP1-WT and GSTP1_4XCys→Ser mutant were not sensitive to NaOH treatment (Fig 3C). These data suggest that GSTP1 is modified by palmitate through a non-ester bond, possibly through an amide bond, which is resistant to NaOH.

A)

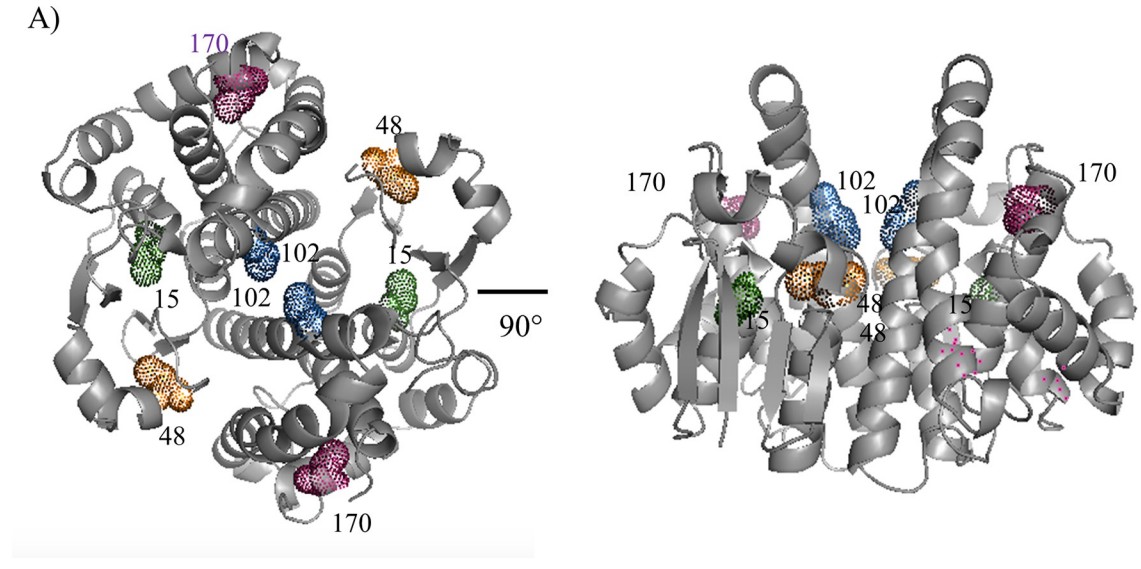

B)

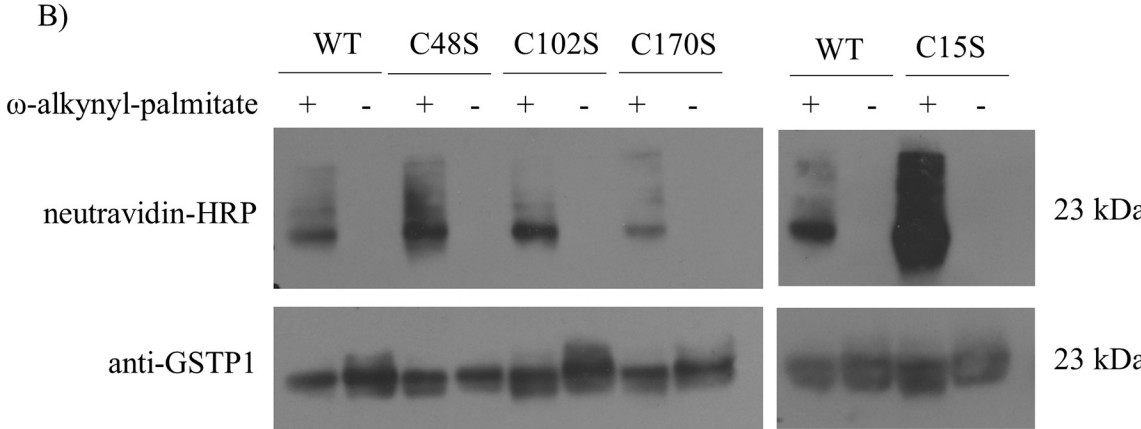

C)

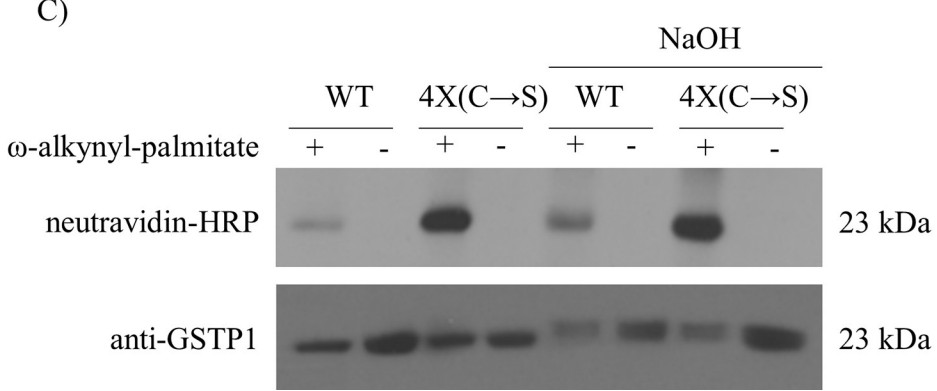

Fig 3. Palmitoylation status of human GSTP1 after mutation of Cys residues to Ser or Ala. (A) Visualization of Cys residues present on GSTP1 dimer were identified in the PDB:3GSS crystal structure and visualized in Pymol; Cys15 (green), Cys48 (orange), Cys102 (blue) and Cys170 (pink). (B) pcDNA3.1-V5_GSTP1-WT, -Cys15Ser, -Cys48Ser, -Cys102Ser or-Cys170Ser were transiently transfected in HEK293T cells, metabolically labelled with ω-alkynyl-palmitate (+) or treated with DMSO (-), immunoprecipiated with mouse anti-V5 antibody (V5-10) conjugated to agarose beads and subjected to click chemistry. Samples were resolved on 11% SDS-PAGE and

transferred to a PVDF membrane. (*Top panel*) Palmitoylated GSTP1 was detected by neutravidin-HRP (1:20000) and (*bottom panel*) total GSTP1 by rabbit pAb anti-GSTP1 (1:5000). **(C)** MCF7 cells stably expressing V5_GSTP1-WT and V5_GSTP1 with all four Cys residues mutated to Ser (Cys15/48/102/170Ser) (V5_GSTP1-4X_(C→S)) were metabolically labelled with ω-alkynyl-palmitate (+) or treated with DMSO (-) for 24–48 h, immunoprecipitated with agarose beads conjugated to mouse anti-V5 antibody (V5-10), and subjected to click chemistry. One half of each sample was treated with NaOH (0.1 M) for one h, the other half was kept at neutral pH. Samples were resolved on 11% SDS-PAGE and transferred to a PVDF membrane. (*Top panel*) Neutravidin-HRP (1:20000) and (*bottom panel*) pAb rabbit anti-GSTP1-GS72 (1:5000) were used to detect palmitoylated and total GSTP1, respectively. Similar results were obtained for an additional independent experiment.

## GSTP1 can be palmitoylated in the absence of a PAT in vitro and this process is Coenzyme A (CoA)-dependent

Due to the atypical nature of GSTP1 palmitoylation through at least one non-ester bond it was impractical to take a traditional site-directed mutagenesis approach for determining which residues are palmitoylated. Thus, a strategy to map sites on GSTP1 by analysing peptide fragments using LC/MS/MS was developed. Initially, we took the simplest approach and labelled recombinant human GSTP1 purified from *E. coli* with palmitoyl-CoA *in vitro*. Prior to mass spectrometry analysis, GSTP1 palmitoylated *in vitro* was biochemically characterized.

CoA is a co-factor in the synthesis and oxidation of fatty acids. It forms a thioester bond with the carboxylic acid of fatty acids; thus acting as an acyl carrier. Esterification of a fatty acid allows the transfer of the fatty acid to another available sulfhydryl group linked by a covalent bond. To investigate if GSTP1 can be palmitoylated *in vitro* in the absence of a PAT, and if the association of GSTP1 with palmitate is due to a covalent modification (rather than a hydrophobic interaction), human GSTP1 purified from *E.coli* was labelled *in vitro* with ω-alkynyl-palmitate, CoA alone, or ω-alkynyl-palmitoyl-CoA, at physiological pH and room temperature. To detect the modification, the fatty acid was "clicked" with azido-biotin and analyzed by western blotting with neutravidin-HRP and rabbit pAb anti-GSTP1 (GS72). A signal for neutravidin was detected for ω-alkynyl-palmitoyl-CoA labelled GSTP1 (Fig 4A, lane 1). To determine if one or more Cys residue(s) are involved in the ω-alkynyl-palmitoyl-CoA labelling, GSTP1 was pre-treated with NEM at pH 7.0 to block Cys residues prior to treatment with ω-alkynyl-palmitoyl-CoA. This resulted in a large drop in neutravidin signal suggesting that one or more Cys residues are either directly modified by palmitate or required for modification of another amino acid, at least *in vitro* (Fig 4A, lane 2). The palmitoylation signal for GSTP1 treated with ω-alkynyl-palmitate without CoA was much weaker than for ω-alkynyl-palmitoyl-CoA (Fig 4A lane 3 versus lane 1) and was reduced by pre-treatment with NEM (Fig 4A, lane 3 versus lane 4). As expected, samples prepared in the presence of CoA alone (with and without NEM) did not have a neutravidin signal (Fig 4A, lanes 5 and 6).

To further characterize the type of bond formed between palmitate and GSTP1, NEM was used to block free Cys residues at neutral pH, and Cys plus Lys residues at the alkali pH 10; *in vitro* palmitoylation of GSTP1was then compared with non-NEM treated controls. Purified GSTP1 was treated with ω-alkynyl-palmitoyl CoA at physiological pH and consistent with Fig 4A, a strong neutravidin signal was detected which was reduced when GSTP1 was pretreated with NEM at pH 7 (Fig 4B, lane 1 versus lane 2). Pretreatment of GSTP1 with NEM at pH 10 resulted in a further reduction of neutravidin signal (Fig 4B, lane 3 versus lane 2), suggesting that Cys and Lys residues could be palmitoylation sites. CoA was used as a negative control and did not show any signal for palmitoylation (Fig 4B, lane 4).

Human GSTP1 purified from *E. coli* was labelled with ω-alkynyl-palmitoyl CoA and incubated with NaOH (0.1 M) for one h, then subjected to click chemistry. Consistent with data from MCF7-V5_GSTP1 cells metabolically labelled with ω-alkynyl-palmitate (Fig 3C), GSTP1 palmitoylated *in vitro* was resistant to NaOH treatment, providing further support for the involvement of an amide bond (Fig 4C).

A)

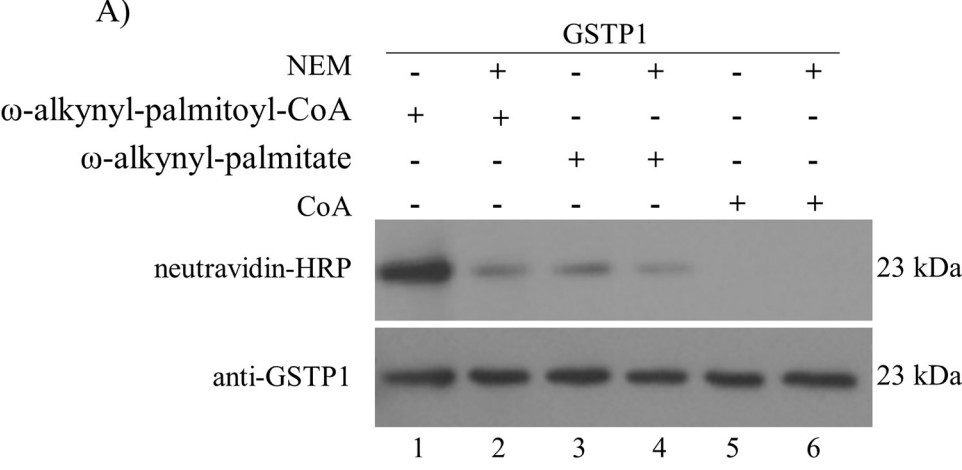

B)

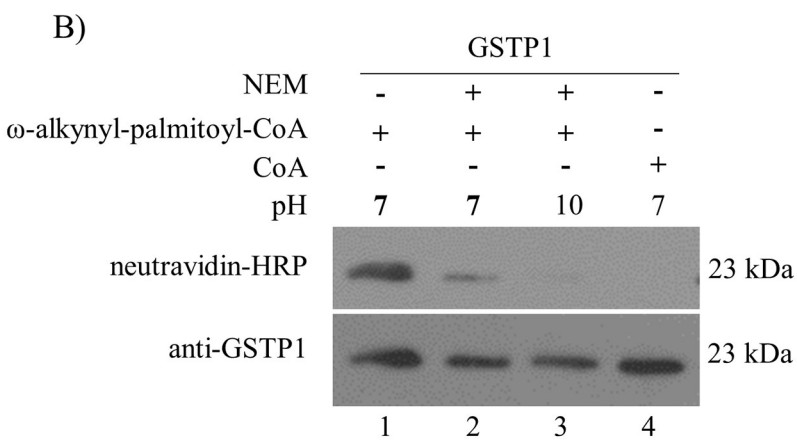

C)

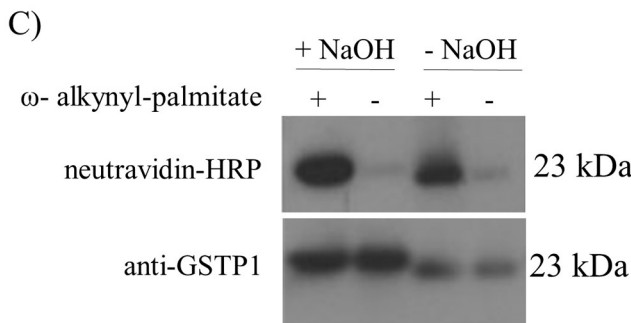

**Fig 4. Biochemical characterization of PAT-independent palmitoylation of human GSTP1 purified from E.coli.** GSTP1 was subjected to a series of chemical treatments pre- or post-palmitoylation *in vitro* and resolved on an 11% SDS-PAGE gel and transferred to a PVDF membrane. Palmitoylated GSTP1 was visualized with neutravidin-HRP (1:20000) and total GSTP1 detected with rabbit pAb anti-GSTP1 (1:1000). **(A)** GSTP1 was incubated in the presence or absence of NEM with ω-alkynyl-palmitoyl-CoA (lanes 1 and 2, respectively), ω-alkynyl-palmitate (lanes 3 and 4) or CoA (lanes 5 and 6). **(B)** Cys and Lys residues were blocked by incubating purified GSTP1 with NEM (10 mM) at pH 7 (lane 2) and pH 10 (lane 3), respectively, then *in vitro* palmitoylation using ω-alkynyl-palmitoyl-CoA was done. **(C)** *In vitro* palmitoylated GSTP1 was treated with NaOH (0.1 M) or MOPS at pH 7 for 1 h and then analyzed. Shown are representative blots and similar results were obtained in at least one additional independent experiment.

### Identification of in vitro palmitoylated GSTP1 residues by LC-MS/MS

Identification of a hydrophobic component, such as palmitate, bound to a hydrophilic peptide is analytically challenging because fatty acids are likely to be lost in the mobile phase or trapped on the reverse-phase column. In addition, when the bond between the protein and the fatty acid is an ester, the chances of detecting the modification are even lower when compared with a more stable amide bond. A strategy to analyze fatty acylated peptides by MS was previously developed [50], and was used to analyze recombinant human GSTP1 purified from *E.coli*. GSTP1 was labelled *in vitro* with palmitoyl-CoA or CoA then digested into peptides with endoproteinase Asp-N or trypsin and analyzed by LC-MS/MS. The following palmitoylated peptides were identified in the palmitoyl-CoA labelled GSTP1: $^{25}$DQGQSWKEEVVTVETWQEGSKAS$^{48}$C$_{palm}$LYGQLPKFQ$^{57}$ (Fig 5A), a single peptide with both Cys102 and Lys103 palmitoylated $^{102}$C$_{palm}$$^{103}$K$_{palm}$YISLIY TNYEAGKDDYVK$^{121}$ (Fig 5B), and $^{102}$C$_{palm}$KYISLIY TNYEAGKDDYVK$^{121}$ (Fig 5C). No evidence of palmitoylated GSTP1 was found for the CoA labelled control.

GSTP1 co-crystallized with ethacrynic acid-GSH [PDB:3GSS, [51]] was analyzed using Pymol to determine the spatial distribution of the *in vitro* palmitoylated residues (Fig 5C). Cys48 (orange), Cys102 (blue) and Lys103 (red) are distributed along the dimer interface [52, 53].

The conservation of these three amino acids, for different GSTP1 orthologs is shown in Table 1 and for different GST isoforms in Table 2. Lys103 is completely conserved in GSTP1 for all species examined. Similarly, Cys48 is highly conserved with the exception of conversion to an Ala in *Xenopus laevis* GSTP1. Cys102 is conserved in all species compared except for mouse, rabbit, and *Xenopus laevis*. The conservation of all three amino acids was low for other human GST isoforms (Table 2).

### Metabolic labelling of MCF7 cells expressing GSTP1_4XCys→Ala/ Lys103Arg mutant retains palmitoylation

Our LC-MS/MS results showed that GSTP1 palmitoylation *in vitro* can occur on at least amino acids Cys48, Cys102, and Lys103, and these sites are candidates for palmitoylation in cells. To determine if cellular palmitoylation of GSTP1 was lost if these residues were no longer available for palmitoylation, the V5_GSTP1_4XCys→Ala/Lys103Arg mutant construct was generated. This was done in the Cys-less construct because even though mass spectrometry analysis (Fig 5) did not identify the Cys15 or Cys170 as palmitoylated sites, the potential of these sites to be modified still exists. It is possible that peptides containing palmitoylated Cys15 and Cys170 could have been missed in our analysis for a few reasons. For example, hydrophobic peptides could have been retained on the column or lost during sample preparation (due to instability). Furthermore, the Cys15 and Cys170 residues could require a PAT for palmitoylation, and therefore would only be modified by palmitate within cells, and not *in vitro*.

The V5_GSTP1_4XCys→Ala/Lys103Arg mutant was stably expressed in MCF7 cells, subjected to metabolic labelling followed by click chemistry and then analyzed by western blotting with neutravidin-HRP and rabbit pAb anti-GSTP1. Expression of V5_GSTP1_4XCys→Ala/Lys103Arg mutant in MCF7 was ~80% lower than V5_GSTP1-WT (Fig 6A, bottom panel). Palmitoylation of V5_GSTP1_4XCys→Ala/Lys103Arg was retained (Fig 6A), and when the palmitoylation (neutravidin) signal was normalized for GSTP1 level, palmitoylation was reduced by only ~20% (Fig 6B).

### Less GSTP1 is associated with the cellular membrane than the cytosolic fraction and palmitoylated GSTP1 is present in both fractions

To quantify the relative level of GSTP1 in the cytosol compared with cellular membranes, subcellular fractions of MCF7-GSTP1 cells were isolated and analyzed by western blotting. GSTP1

A)

$$DQGQSWKEEVVTVETWQEGSLKAS^{48}C_{palm}LYGQLPKFQ$$

Endoproteinase Asp-N digestion

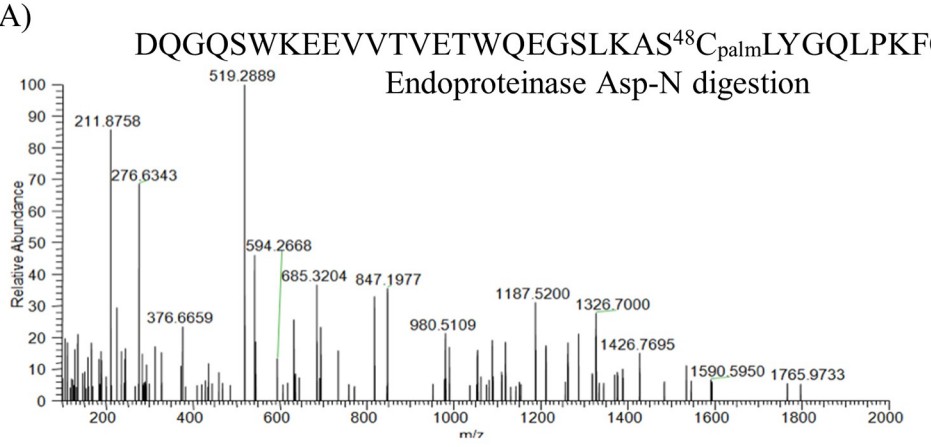

B)

$$^{102}C_{palm}{}^{103}K_{palm}YISLIYTNYEAGKDDYVK$$

Trypsin digestion

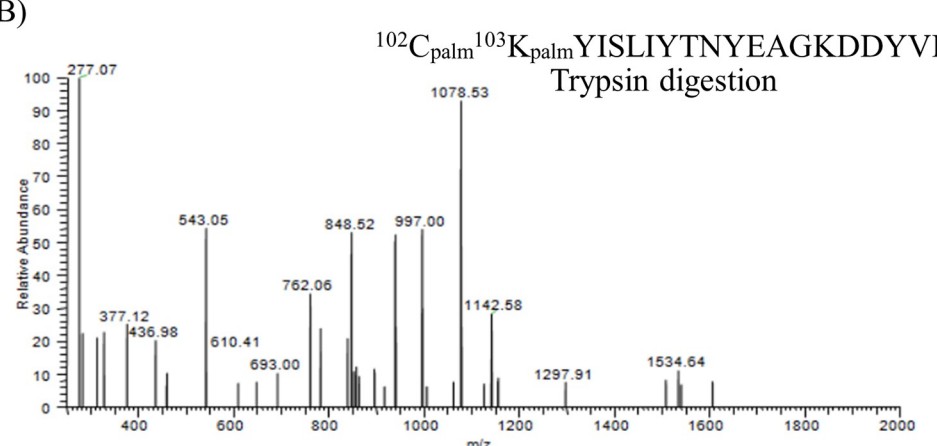

C)

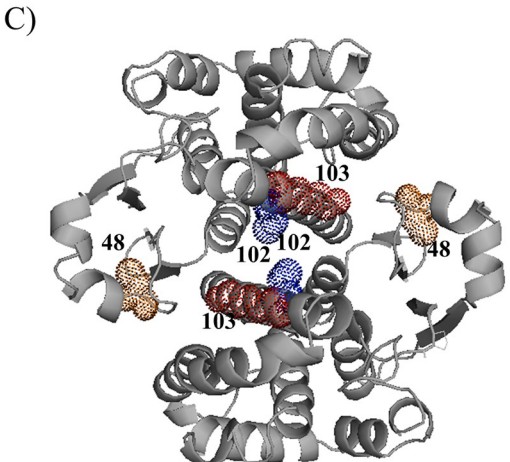

**Fig 5. Mapping of palmitoylated amino acids on in vitro palmitoylated GSTP1 by LC-MS/MS.** Human GSTP1 purified from *E.coli* was palmitoylated *in vitro* and digested with proteases into peptides. After enriching for palmitoylated peptides, the samples were analyzed by LC-MS/MS. Shown are the annotated MS/MS spectra of GSTP1 peptides covering $^{48}C_{palm}$, $^{102}C_{palm}$, and $^{103}K_{palm}$, with their respective backbone cleavage maps. **(A)** $^{48}C_{palm}$-containing GSTP1 peptide obtained by Asp-N digestion, **(B)** tryptic GSTP1 peptides with $^{102}C$ and $^{103}K$ palmitoylation. **(C)** Spatial distribution of *in vitro* palmitoylated Cys48 (orange), Cys102 (blue), and Lys103 (red) of GSTP1 identified by LC-MS/MS was visualized in Pymol using the crystal structure (PDB 3GSS).

**Table 1. Conservation of the *in vitro* palmitoylated amino acids among different GSTP1 orthologues.** Residue number of the human GSTP1 is being used as reference.

| Species | Cys48 | Cys102 | Lys103 |
|---|---|---|---|
| Human | Cys | Cys | Lys |
| Mouse 1 | Cys | Gly | Lys |
| Mouse 2 | Cys | Gly | Lys |
| Rat | Cys | Cys | Lys |
| Chimpanzee | Cys | Cys | Lys |
| Gorilla | Cys | Cys | Lys |
| Rabbit | Cys | Ile | Lys |
| Pig | Cys | Cys | Lys |
| Frog | Ala | Gln | Lys |

was detected in the total cell homogenate (input), cytosolic, and membrane fractions (Fig 7A). To compare GSTP1 levels in the membrane and cytosolic fractions, bands were quantified using imageJ and plotted relative to the level in the cytosolic fraction (a total protein stain confirmed equal loading). The cellular membrane pellet had 64% less GSTP1 than the cytosolic fraction (Fig 7B).

To identify the subcellular fraction in which palmitoylated GSTP1 resides, MCF7-V5_GSTP1 cells were metabolically labelled with ω-alkynyl-palmitate and cytosolic and total cellular membrane fractions isolated. GSTP1 was then immunoprecipitated using agarose beads conjugated to mouse anti-V5 antibody directly from the cytosolic fraction, while the membrane fraction was first solubilized using SDS (0.1%). Samples were then subjected to click chemistry. Samples were resolved on SDS-PAGE and transferred to a PVDF membrane. GSTP1 palmitoylation and total GSTP1 were detected with neutravidin-HRP or rabbit pAb anti-GSTP1, respectively. Palmitoylated GSTP1 was present in the membrane and the cytosolic fraction (Fig 7C). The differences in conditions for immunoprecipitation from

**Table 2. Conservation of the *in vitro* palmitoylated amino acids among different human GST isoforms.** Residue number of the human GSTP1 is being used as reference.

| Species | Cys48 | Cys102 | Lys103 |
|---|---|---|---|
| GSTP1 | Cys | Cys | Lys |
| GSTM1 | Leu | Met | Gln |
| GSTM2 | Leu | Met | Gln |
| GSTM3 | Leu | Thr | Gln |
| GSTM4 | Leu | Asn | Gln |
| GSTM5 | Leu | Met | Glu |
| GSTA1 | Leu | Glu | Met |
| GSTA2 | Leu | Glu | Met |
| GSTA3 | Leu | Glu | Met |
| GSTA4 | Leu | Glu | Leu |
| GSTA5 | Leu | Glu | Met |
| GSTO1 | Asn | Ser | - |
| GSTO2 | His | His | - |
| GSTS | Leu | Ser | Cys |
| GSTT1 | Asn | Arg | Ser |
| GSTT2 | Asn | Arg | Thr |

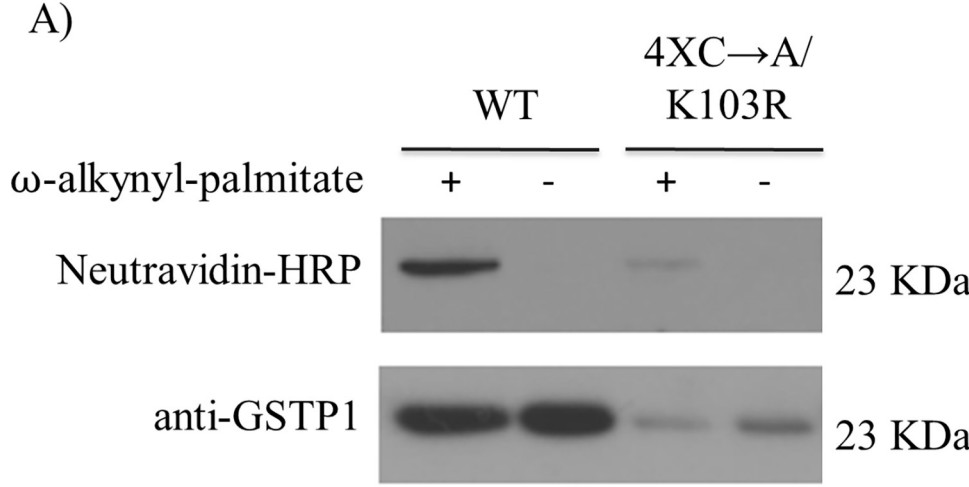

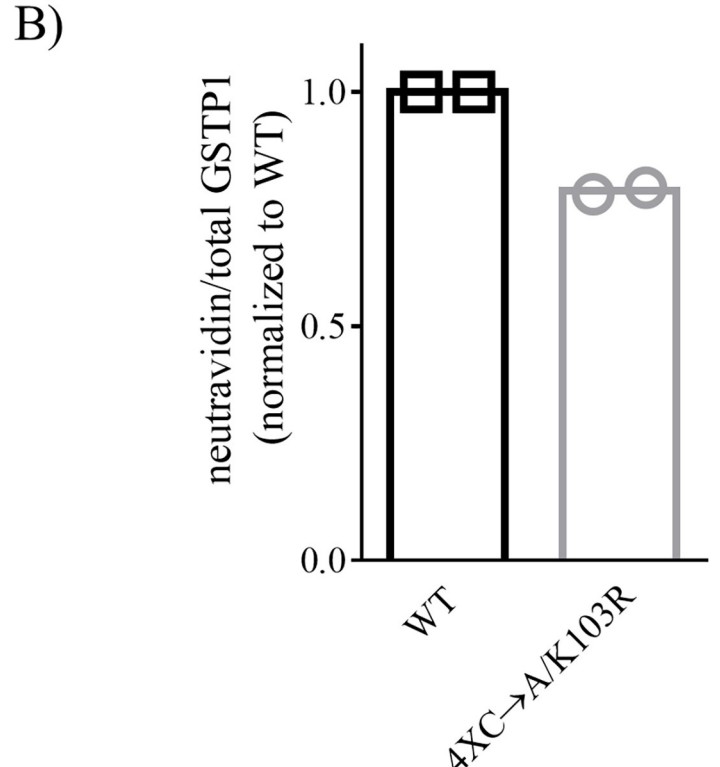

**Fig 6. Characterization of V5_GSTP1_4XCys→Ala/Lys103Arg palmitoylation. (A)** The V5_GSTP1_4XCys→Ala/
Lys103Arg mutant was generated and stably expressed in MCF7 cells. This cell line and MCF7-V5_GSTP1-WT were
then metabolically labelled with ω-alkynyl-palmitate (+) or treated with DMSO (-) for >24 h. Cells were lysed and
GSTP1 immunoprecipitated with agarose beads conjugated to mouse anti-V5 antibody (V5-10), subjected to click
chemistry, resolved on 11% SDS-PAGE and transferred to a PVDF membrane. Palmitoylated GSTP1 was visualized
with neutravidin-HRP (1:20000) and total GSTP1 with rabbit pAb anti-GSTP1 (1:5000). **(B)** Densitometry on
neutravidin and anti-GSTP1 was performed using ImageJ software. The palmitoylated GSTP1 (neutravidin-HRP) was
divided by the total GSTP1 signal and plotted normalized to WT-GSTP1. Two independent experiments were
completed, bars represent means and symbols (□, ○) represent individual experiments.

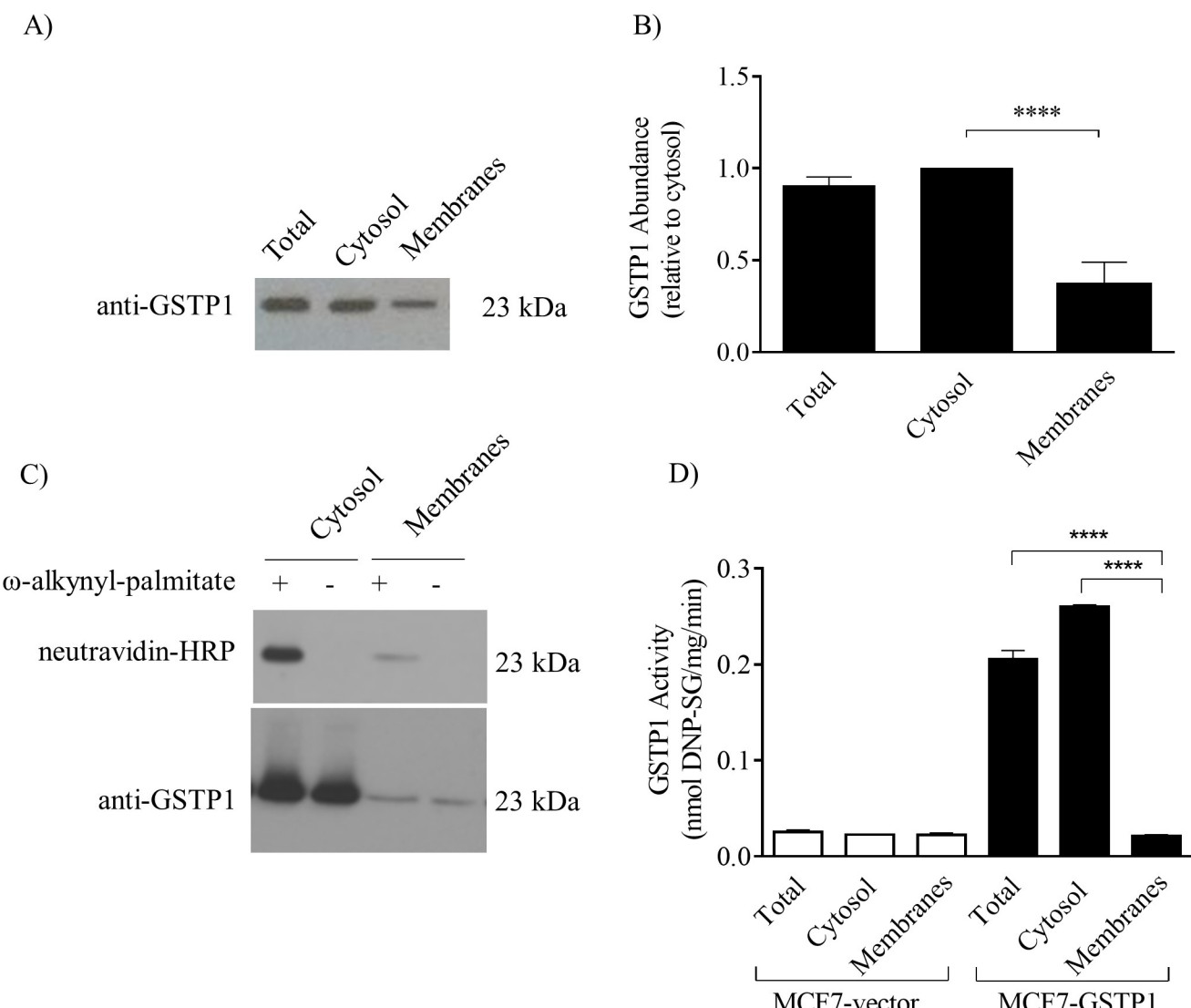

**Fig 7. Characterization of GSTP1 palmitoylation and activity from total, cytosolic and cellular membrane fractions of MCF7 cells.**
MCF7-V5_GSTP1, MCF7-GSTP1, or MCF7-vector cells were disrupted by $N_2$ cavitation, nuclei and unbroken cells were removed by centrifuging at 2000 *g*. Supernatant (total) was collected and cytosolic fraction was separated from cellular membranes by centrifugation at 100000 *g* for 1 h. **(A)** 10 μg of protein sample from total input, cytosol, and membrane fractions from MCF7-GSTP1 cells were resolved on 11% SDS-PAGE and tranferred to a PVDF membrane. Total GSTP1 was detected with rabbit pAb anti-GSTP1 (1:5000) and equal protein loading was confirmed with amido black staining. **(B)** Densitometry of anti-GSTP1 in panel A was performed using ImageJ software. The level of GSTP1 from each fraction relative to cytosolic was plotted with bars representing means (±SD) from 5 independent experiments. Significant difference is indicated with **** $p < 0.0001$ (one-way ANOVA with Dunnett multiple comparisons test). **(C)** GSTP1 was immunoprecipitated from cytosolic and membrane fractions of MCF7-V5_GSTP1 cells metabolically labelled with ω-alkynyl palmitate (+) or DMSO containing medium (—) using agarose beads conjugated to mouse anti-V5 antibody (V5-10), and subjected to click chemistry, resolved on a 11% SDS-PAGE and tranferred onto a PVDF membrane. (*top panel*) Palmitoylated GSTP1 was detected with neutravidin-HRP (1:20000) and (*bottom panel*) total GSTP1 with rabbit pAb anti-GSTP1 (1:5000). Palmitoylated GSTP1 from cytosolic and membrane fractions (neutravidin-HRP) and total GSTP1(anti-GSTP1) from MCF7-V5_GSTP1 are shown. Shown is a representative blot and similar results were found in one additional experiment. **(D)** GSTP1 activity of total cell lysate, cytosol and membrane fraction from MCF7-vector and MCF7-GSTP1. Three μg of total cell lysate, cytosol, and membranes were incubated with 1 mM CDNB and 2.5 mM GSH. Formation of the DNP-GS conjugate was measured at 340 nm. Abssorbance values were normalized for blanks and spontaneous formation of DNP-GS in the absence of cellular fraction. Rates were normalized for relative GSTP1 levels estimated by densitometry, with bars representing means (±SD) of three independent experiments. Significant differences compared to MCF7-GSTP1 membranes are indicated with **** $p \leq 0.0001$, (one-way ANOVA with Dunnett multiple comparisons test).

the cytosolic versus the membrane fractions prevents comparison of relative levels of total and palmitoylated GSTP1 between the cytosol and membranes.

## GSTP1 associated with cell membranes does not catalyze CDNB conjugation with GSH

CDNB is well characterized as a substrate for GSTs and GSTP1 catalyzes its conversion to DNP-GS [3]. Although GSTP1 strongly associates with the plasma membrane [24], the influence of this localization on GSTP1 function was not previously characterized. The rate of DNP-GS formation by GSTP1 in the total cell lysate (total), cytosolic fraction, and total cellular membranes (membranes) fraction was determined. The rate of DNP-GS formation for the cytosolic fraction from MCF7-GSTP1 was 0.27 nmol of DNP-GS/mg protein/min and not significantly different from the total cell lysate of MCF7-GSTP1 at 0.23 nmol of DNP-GS/mg protein/min (Fig 7D). In contrast, GSTP1 activity in the membrane fraction of MCF7-GSTP1 was only 0.02 nmol of DNP-GS/mg protein/min which was not significantly different from the formation of DNP-GS by total cell lysate and subcellular fractions isolated from MCF7-vector cells (negative control) (Fig 7D). GSTP1 in the membrane fraction did not demonstrate any activity even after normalization for GSTP1 levels, suggesting that the GSTP1 associated with the membranes is not catalytically active (at least for CDNB) (Fig 7D).

## Purified GSTP1 associates with the MCF7 plasma-membrane enriched fraction

Next, experiments were designed to determine if palmitoylation influences the ability of GSTP1 to associate with cellular membranes. Human GSTP1 purified from *E. coli* or MCF7-GSTP1 cells grown under conditions with regular FBS (GSTP1-FBS) or charcoal stripped FAF-FBS (GSTP1-FAF), were subjected to *in vitro* palmitoylation and incubated with plasma membrane-enriched fractions isolated from untransfected MCF7 cells. Samples were pelleted by ultracentrifugation and analyzed by immunoblotting. GSTP1 was detected with rabbit pAb anti-GSTP1 and the rabbit pAb anti-Na$^+$/K$^+$-ATPase was used as a loading control for plasma membrane-enriched fractions. GSTP1 is not expressed endogenously in MCF7 cells, therefore the plasma membrane enriched fraction alone did not have any detectable signal for GSTP1 (Fig 8, left lane in all blots). GSTP1 from *E. coli* (Fig 8A and 8B), as well as MCF7 cells GSTP1-FBS (Fig 8C and 8D) and GSTP1-FAF (Fig 8E and 8F) associated with the plasma membrane-enriched fraction in the absence of *in vitro* palmitoylation. After *in vitro* palmitoylation, no significant increase in the amount of purified GSTP1 interacting with the MCF7 plasma membrane enriched fraction was detected (Fig 8A–8E, right hand lane and column). Although there was a trend for human GSTP1 purified from *E. coli* (p = 0.2, Fig 8B) and MCF7-GSTP1 grown under FAF conditions (p = 0.2, Fig 8F), neither were significant. Thus, before and after *in vitro* palmitoylation, GSTP1 is capable of associating with cellular membranes.

## Discussion

We report for the first time that GSTP1, a protein with pleiotropic functions and multiple cellular localizations, is modified by palmitate. Palmitoylation can result in dynamic changes to protein stability, folding, function, cellular localization, as well as interactions with other proteins [29, 30]. The knowledge gained from this work lays the foundation for understanding how palmitoylation contributes to the diverse functions of GSTP1.

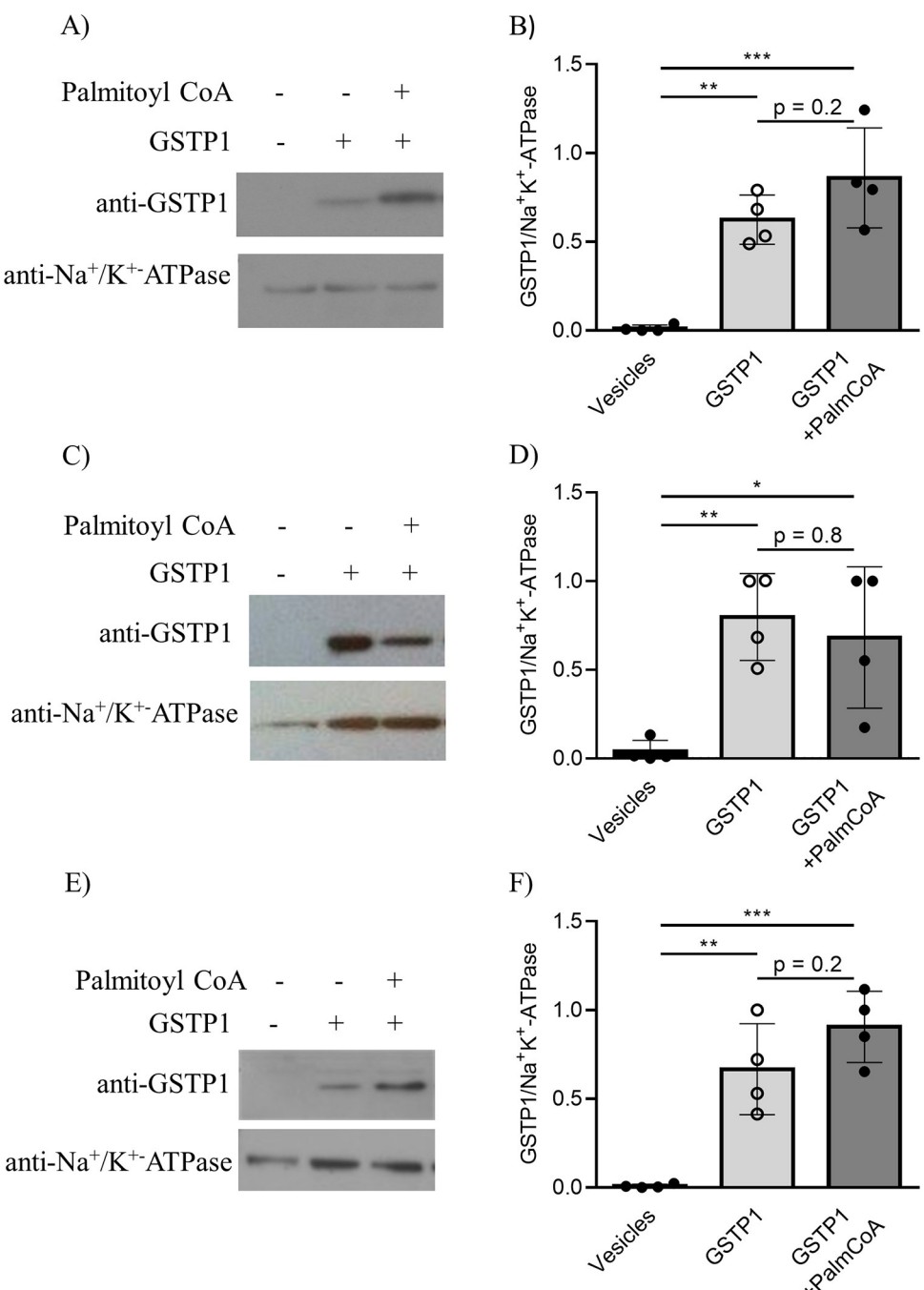

**Fig 8. Association of purifed GSTP1 with the MCF7 plasma membrane-enriched fraction.** Human GSTP1 grown in and purified from **(A, B)** *E. coli* or **(C-F)** MCF7-GSTP1 cells grown in the presence of **(C, D)** regular FBS or **(E, F)** charcoal stripped fatty acid free FBS were tested for their ability to interact with the MCF7 plasma membrane-enriched fraction. GSTP1 from the different sources were left untreated or *in vitro* labelled with ω-alkynyl-palmitoyl-CoA. Untreated and *in vitro* palmitoylated GSTP1 (2 μg) were incubated with the MCF7 plasma membrane-enriched fraction (20 μg) for 30 min at 37°C. The MCF7 plasma membrane-enriched fraction without GSTP1 were incubated under identical conditions. Cellular membranes were pelleted by ultracentrifugation at 100000 *g* for 1 h, washed with PBS 1X + Triton-X-100 0.1%, resolved by 11% SDS-PAGE, and transferred onto a PVDF membrane. Membranes were immunoblotted for GSTP1 using the pAb rabbit anti-GSTP1(GS72) (1:5000). Equality of plasma membrane fraction loading was determined by probing for the integral plasma membrane protein $Na^+/K^+$-ATPase using the rabbit pAb anti- $Na^+/K^+$-ATPase (1:10000). **(A, C, E)** A representative blot is shown for each source of GSTP1 and similar results were observed in at least three additional independent experiments. **(B, D, F)** Densitometry analysis of GSTP1 and $Na^+/K^+$-ATPase bands was performed with ImageJ software. GSTP1 associated with the membrane was normalized

for plamsa membrane fraction protein loading using the $Na^+$/$K^+$-ATPase signal. Bars represent the mean ± SD of 4 independent experiments with data points from individual determinations shown. Signficant differences are indicated with * $p<0.05$, ** $p<0.01$, *** $p<0.001$ and non-significant differences are indicated with a p-value, (one-way ANOVA followed by a Tukey multiple comparisons test).

GSTP1 was identified as a palmitoylated protein after 8 h of metabolic labelling with ω-alkynyl-palmitate, while palmitoylation of other proteins (e.g., N-Ras) is very rapid (less than 1 h) and standard protocols are 4 h [45]. The longer metabolic labelling required for detection of GSTP1 palmitoylation could suggest that GSTP1 is newly synthesized prior to incorporation of ω-alkynyl-palmitate (possibly palmitoylation occurs co-translationally). Consistent with this, the half-life of GSTP1 in cells can be up to 9 h, depending on the cell type [54]. These data suggest GSTP1 palmitoylation is more stable and less dynamic than other palmitoylated proteins. Modification of GSTP1 by palmitate through an amide bond and/or at a solvent inaccessible site would be consistent with less dynamic palmitoylation.

Although proteins typically undergo *S*-palmitoylation, individual Cys (GSTP1-Cys15Ser, -Cys48Ser, -Cys102Ser, and -Cys170Ser) and Cys-less GSTP1 (V5_GSTP1_4XCys→ Ser) over-expressed in MCF7 retained palmitoylation after metabolic labelling with ω-alkynyl palmitate, demonstrating that GSTP1 Cys residues are not the only requirement for palmitoylation. Indeed, the individual Cys mutants and Cy-less GSTP1 retained strong levels of palmitoylation, demonstrating that palmitoylation can occur at non-Cys sites. Treatment of palmitoylated GSTP1-WT and the Cys-less mutant with a strong base (NaOH) did not cleave palmitate. Thio and oxy-ester bonds are cleaved by strong base, suggesting that GSTP1 is palmitoylated through an amide bond. Due to the dynamic nature of *S*-palmitoylation, these data do not rule out the possibility that GSTP1 Cys residues are utilized and occupancy of Cys residues could be transient. Thus, we conclude that GSTP1 is palmitoylated on at least one non-Cys residue, likely through an amide bond, potentially at a Lys residue.

Palmitoylation at non-Cys sites could potentially occur at one or more of the twelve Lys, ten Ser, or nine Thr residues contained in GSTP1. Due to the number of potential sites and the possibility of palmitoylation occurring at multiple sites, a traditional site-directed mutagenesis approach to map the location of palmitoylation was not feasible. *In vitro* palmitoylation of GSTP1 was used as a strategy to set up conditions for sequencing of GSTP1 peptides for palmitoylated amino acids by LC-MS/MS. To begin with, *in vitro* palmitoylation of GSTP1 was characterized. Furthermore, we found that GSTP1 is palmitoylated *in vitro* and in the absence of a PAT, after treatment with ω-alkynyl-palmitoyl-CoA, but not ω-alkynyl-palmitate suggesting that a covalent modification occurs (not just hydrophobic binding). *In vitro* palmitoylation of GSTP1 was markedly reduced when the protein was treated with NEM at pH 7 (blocks Cys) before subjecting GSTP1 to *in vitro* palmitoylation and further reduced when treated with NEM at pH 10 (blocks Cys and Lys), suggesting that *in vitro* palmitoylation occurs through a thioester and amide bond or that Cys residues are required for the acylation onto a final Lys acceptor (e.g., Lys103).

Although application of MS to lipidomic research is technically challenging, we were able to apply a protocol optimized to analyze peptides modified by fatty acids [50]. MS analysis of *in vitro* palmitoyl-CoA labelled GSTP1 identified Cys48, Cys102, and Lys103 as amino acid residues modified by palmitate through a PAT-independent process, at least *in vitro*. Cys48 is located in the $NH_2$-terminal half of GSTP1 whereas Cys102 is at the dimer interface [53]. Interestingly, both of these sites are known to be important for NO binding and *S*-glutathionylation [14, 55], and because of the nature of Cys residues, a fine-tuned combination of multiple co- or post-translational modifications of these sites (including *S*-palmitoylation) could be

important for regulating the multiple functions of GSTP1. Non-PAT mediated palmitoylation of GSTP1 at Cys15 and Cys170 was not detected, which does not exclude these residues from being substrates for PAT-mediated palmitoylation within the cells (nor Cys48, Cys102 or Lys103.). Moreover, Cys15 and Cys170 are solvent accessible, suggesting that palmitate could have been lost throughout the '*in vitro*' process. When corrected for GSTP1 levels V5_GSTP1_4XCys→Ala/Lys103Arg retained a similar palmitoylation signal as GSTP1-WT. These data could suggest that there are palmitoylation sites within the cells in addition to or different from Cys48, Cys102, and Lys103. Analysis of *in vitro* labelled GSTP1 allows the identification of only PAT-independent palmitoylated residues, the presence of different PATs within cells likely allows acylation of additional sites.

Fatty acid modification that results in amide bond formation has been reported previously. For example, the $NH_2$-terminal position of the Cys residue of the Sonic Hedgehog is palmitoylated by hedgehog acyltransferase (HHAT) [56, 57]. Aquaporin 0, a transmembrane protein, can also be acylated on an internal Lys by oleic acid [58]. It was surprising that a stable amide bond was formed *in vitro*, at neutral pH, without the requirement of an enzyme. The presence of the singly palmitoylated peptide with only $^{102}C_{palm}$ and the absence of a singly palmitoylated peptide with only $^{103}K_{palm}$ suggest that $^{102}C$ palmitoylation precedes (or even promotes) $^{103}K$ palmitoylation.

*S*-palmitoylation typically has a cycle of palmitoylation and de-palmitoylation regulated by PATs and thioesterases, respectively [59]. This process permits dynamic localization and the protein is commonly palmitoylated while in the membrane and depalmitoylated while in the cytosol. In contrast, subcellular fractionation analysis showed that not only membrane associated but also cytosolic GSTP1 is palmitoylated, suggesting that palmitoylation of cytosolic GSTP1 is not restricted to membrane interaction. Palmitoylation could be utilized to stabilize the quaternary or tertiary structure of GSTP1, for dimerization, oligomerization or association with other proteins. Curiously, Cys48, Cys102, and Lys103, which were shown to be palmitoylated *in vitro*, are present in the hydrophobic interface of the dimer, where a 'lock and key' motif is formed by the Tyr50 in the $NH_2$-terminus of one monomer, which wedges into a hydrophobic pocket in the COOH-terminus of the other monomer (amino acids 80 to 110) [53]. Palmitoylation of amino acids in this vicinity could increase the hydrophobicity of the pocket and help stabilize the dimer.

Identifying palmitoylated GSTP1 in the membrane and cytosolic fractions is consistent with palmitoylation occurring at multiple amino acids. Interestingly, in contrast with cytosolic, the membrane-associated GSTP1 had no detectable catalysis of CDNB conjugation with GSH. This could imply that GSTP1 associated with cellular membranes is catalytically inactive and plays a different role at this location (e.g., a signaling rather than catalytic function). Lastly, purified GSTP1 was found to associate with plasma membrane-enriched fractions isolated from MCF7 cells, however, there was no significant difference between GSTP1 subjected to *in vitro* palmitoylation and untreated GSTP1. The reasons for this are unknown, but suggest the role of palmitoylation is more complex than simply localizing GSTP1 to the plasma membrane.

GSTP1 is a multi-faceted protein with important functions including detoxification, cell signalling, redox balance, *S*-glutathionylation, and NO regulation. These diverse functions along with multiple cellular localizations suggest that GSTP1 is differentially regulated throughout the cell. The conclusions that could be drawn from this research were limited by complex multi-site palmitoylation, preventing mutagenesis studies for comparison of palmitoylated and non-palmitoylated forms to identify the function(s) of GSTP1 palmitoylation. However, the novel finding that GSTP1 is palmitoylated lays the foundation for future studies to understand how lipidation may influence GSTP1 function. Given the importance of GSTP1

in cell biology, cancer and multiple other pathologies, further study of GSTP1 palmitoylation is warranted.

## Supporting information

**S1 Raw images. Raw images for all immunoblot data in manuscript.**
(PDF)

**S1 File. Raw data from which means ± S.D. were derived for Figs 1, 6–8.**
(PDF)

## Acknowledgments

Dr. Philip G. Board (Australian National University, Canberra) is thanked for the kind gift of the pBluescriptSK(−) GSTP1 construct. Diane Swanlund is thanked for excellent technical assistance.

## Author Contributions

**Conceptualization:** Vanessa Marensi, Yuhuan Ji, Luc G. Berthiaume, Elaine M. Leslie.

**Data curation:** Cheng Lin.

**Formal analysis:** Vanessa Marensi, Yuhuan Ji, Cheng Lin, Elaine M. Leslie.

**Funding acquisition:** Cheng Lin, Luc G. Berthiaume, Elaine M. Leslie.

**Investigation:** Vanessa Marensi, Megan C. Yap, Yuhuan Ji, Cheng Lin, Luc G. Berthiaume, Elaine M. Leslie.

**Methodology:** Vanessa Marensi, Megan C. Yap, Yuhuan Ji, Cheng Lin, Luc G. Berthiaume.

**Project administration:** Elaine M. Leslie.

**Resources:** Cheng Lin, Luc G. Berthiaume, Elaine M. Leslie.

**Software:** Cheng Lin.

**Supervision:** Cheng Lin, Elaine M. Leslie.

**Validation:** Megan C. Yap.

**Visualization:** Megan C. Yap.

**Writing – original draft:** Vanessa Marensi, Elaine M. Leslie.

**Writing – review & editing:** Vanessa Marensi, Megan C. Yap, Yuhuan Ji, Cheng Lin, Luc G. Berthiaume, Elaine M. Leslie.

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
