## [Decision Letter · Decision Letter 0]

10 Jul 2024

PONE-D-24-21632Glutathione transferase P1 is modified by palmitatePLOS ONE

Dear Dr. Leslie,

Thank you for submitting your manuscript to PLOS ONE. After careful consideration, we feel that it has merit but does not fully meet PLOS ONE’s publication criteria as it currently stands. Therefore, we invite you to submit a revised version of the manuscript that addresses the points raised during the review process.

We look forward to receiving your revised manuscript.

Kind regards,

Mohd Akbar Bhat, Ph.D.

Academic Editor

PLOS ONE

Journal Requirements:

We acknowledge the support of the Natural Sciences and Engineering Research Council of Canada (NSERC), [funding reference number RGPIN-2017-06154], the Canadian Institutes of Health Research (CIHR grant MOP-272075), and the US National Institutes of Health (NIH grants P41 GM104603, R24 GM134210). LGB acknowledges the Alberta Cancer Foundation grant #27369. Diane Swanlund is thanked for excellent technical assistance.

Natural Sciences and Engineering Research Council of Canada (NSERC) to EML, [funding reference number RGPIN-2017-06154], https://www.nserc-crsng.gc.ca.

Canadian Institutes of Health Research to EML (CIHR grant MOP-272075) https://cihr-irsc.gc.ca/e

US National Institutes of Health (NIH grants P41 GM104603, R24 GM134210) to CL. 

Alberta Cancer Foundation (grant #27369 to LGB).https://albertacancer.ca

None of the funders played any role in the study design, data collection and analysis, decision to publish, or preparation of the manuscript.

Reviewers' comments:

Reviewer's Responses to Questions

**Comments to the Author**

1. Is the manuscript technically sound, and do the data support the conclusions?

Reviewer #1: Yes

Reviewer #2: Yes

2. Has the statistical analysis been performed appropriately and rigorously? 

Reviewer #1: Yes

Reviewer #2: Yes

3. Have the authors made all data underlying the findings in their manuscript fully available?

Reviewer #1: Yes

Reviewer #2: Yes

4. Is the manuscript presented in an intelligible fashion and written in standard English?

Reviewer #1: Yes

Reviewer #2: No

5. Review Comments to the Author

**Reviewer #1:** Only this sentnce "GSTP1 is the most prevalent and widely distributed GST, in humans GSTP1 is found

in most cell types and tissues, with the exception of healthy adult hepatocytes (4, 5)" may require modification for improved comprehension

**Reviewer #2:** R1:

• enjoyable and modern topic.

• The title is suitable.

• All data are presented in intense and poring way.

• Authors must read previous papers to re-write this paper as it deserve.

Abstract:

-better to divide abstract (summary of your study) to sub titles include:

-background

-aim or objective of the study

-methods

-results

-conclusions of results

-this give complete idea about the research

-enough to use glutathione transferase P1, palmitic acid as key-words

INTRODUCTION:

-authors can define the abbreviation which mentioned for one time, and list the abbreviations that repeated for many time.

RESULTS:

-research performed in highly precise methods, but more than requested data were mentioned.

please re-write this section depend on the following guide to represent your idea. to describe this section please summarize the methods as below in subtitle:

- experiential animal.

- sample collection or extraction.

- sample preparation

- materials(use scientific name and avoid the trade name and marks)

- principle(use standard principle).

- avoid repetition of information.

-please do not use trade mark and names of companies, persons or labs to prove your data source. write in scientific way.

-avoid poring details that describe your way and behavior to perform research.

-Marensi is one of the authors not source of figure.

RESULTS:

-the author results were describe the principle of study sample or methods for sample preparation.

result must contain only the result of authors experiment.

-please do not discuss or justifies the results in this section.

DISCUSSION:

-good discussion of the results. this paragraph to be completed need prove of previous or recent studies that accordance with or different from author results.

-the discussion must be limited to results only.

-please compare these result with previous researches, and so on ..... for all results.

-better to discuss directly the results only in prief, precise and scientific language.

-avoid repetition of data.

-by the end of discussion you can add limitations and recommendation in points.

6. PLOS authors have the option to publish the peer review history of their article (what does this mean?). If published, this will include your full peer review and any attached files.

Reviewer #1: **Yes: **Samuel Acquah

Reviewer #2: No

---

## [Author Response · Author response to Decision Letter 0]

22 Jul 2024

Included in the cover letter which has better formatting.

July 21st, 2024

Dear Dr. Mohd Akbar Bhat 

Thank you for your email of July 10th, 2024 inviting us to submit a revised version of our manuscript PONE-D-24-21632 entitled “Glutathione Transferase P1 is modified by palmitate”. We appreciate the reviews provided. As requested, we have responded to the specific comments (bold and italicized) of the editor and reviewers point by point below, and where necessary, revised the manuscript accordingly. 

Editor comments:

1. When submitting your revision, we need you to address these additional requirements.Please ensure that your manuscript meets PLOS ONE's style requirements, including those for file naming. 

The manuscript has now been formatted according to the PLOS ONE style templates.

2. We note that you have provided funding information that is not currently declared in your Funding Statement. However, funding information should not appear in the Acknowledgments section or other areas of your manuscript. We will only publish funding information present in the Funding Statement section of the online submission form. Please remove any funding-related text from the manuscript and let us know how you would like to update your Funding Statement.

We have removed all funding statements from the manuscript body. We do not need to amend the online submission form, it is complete as is.

The data are only supportive and not core to the manuscript so we have removed the phrase that referred to the data.

4. Please include captions for your Supporting Information files at the end of your manuscript, and update any in-text citations to match accordingly.

We have now included a caption for our single supporting information file that contains original uncropped and unadjusted images underlying all blot or gel results. We have not referred to this file in the text. 

5. PLOS ONE now requires that authors provide the original uncropped and unadjusted images underlying all blot or gel results reported in a submission’s figures or Supporting Information files. 

All of our blot/gel image data for figures contained in the manuscript are in Supporting Information Fig S1. 

Reviewer #1 comments: 

This reviewer had only one suggestion: The sentence "GSTP1 is the most prevalent and widely distributed GST, in humans GSTP1 is found in most cell types and tissues, with the exception of healthy adult hepatocytes (4, 5)" may require modification for improved comprehension.

We have now modified this sentence to: GSTP1 is a prevalent and widely distributed GST, found in most human cell types and tissues, with the exception of healthy adult hepatocytes.

Reviewer #2 comments: 

This reviewer had suggestions about the presentation of the manuscript with no criticisms of the data or research itself. The following positive comments were made:

• enjoyable and modern topic.

• The title is suitable.

• All data are presented in intense and poring way.

Changes requested:

1)The reviewer asked us to divide the abstract into subtitles (Background, Aim, Methods, Results, Conclusions). 

This does not seem to be a formatting requirement of PLOS ONE and in our opinion the abstract flows better without subheadings. It is our preference to leave the abstract as it is. We will comply with the reviewer’s request if the editor prefers.

2) Enough to use glutathione transferase P1, palmitic acid as key-words.

We think it’s important to have the different types of palmitoylation (N-, S- and O-palmitoylation) as keywords so we have kept these (rather than replacing them with palmitic acid). We have deleted glutathione.

3) INTRODUCTION:

-authors can define the abbreviation which mentioned for one time, and list the abbreviations that repeated for many time.

The abbreviation MCF-7 has now been defined as Michigan Cancer Foundation-7 in the text and the abbreviations list. All other abbreviations have been defined with first use. 

The reviewer highlighted the last paragraph of the introduction and stated “results” as a comment. 

Our interpretation of this comment was that the reviewer thinks this section belongs in the results. We have deleted this paragraph and replaced it with the objectives of the study.

4) Methods-research performed in highly precise methods, but more than requested data were mentioned.

please re-write this section depend on the following guide to represent your idea. To describe this section please summarize the methods as below in subtitle:

- experiential animal.

- sample collection or extraction.

- sample preparation

- materials(use scientific name and avoid the trade name and marks)

- principle(use standard principle).

- avoid repetition of information.

We have gone through our methods section and removed repetitive information but we have not subdivided under the suggested headings because we feel they are too general or do not apply (for example we don’t have experimental animals in our study). 

-please do not use trade mark and names of companies, persons or labs to prove your data source. Write in scientific way.

It is necessary to provide names of companies so that people can order comparable reagents to complete similar studies. If there is a PLOS ONE policy that we have missed in the instructions to authors please let us know and we will comply.

-avoid poring details that describe your way and behavior to perform research.

It is not clear what this means.

5) RESULTS:

-the author results were describe the principle of study sample or methods for sample preparation.

result must contain only the result of authors experiment.

-please do not discuss or justifies the results in this section.

We have gone through our results section and deleted any unnecessary discussion.

6) DISCUSSION:

a) Second paragraph good discussion of the results. This paragraph to be completed need prove of previous or recent studies that accordance with or different from author results.

-the discussion must be limited to results only.

Appropriate references to previous studies are included.

b)-please compare these result with previous researches, and so on ….. for all results.

-better to discuss directly the results only in prief, precise and scientific language.

-avoid repetition of data.

-by the end of discussion you can add limitations and recommendation in points.

Where possible we have shortened the discussion and major limitations are now described in two sentences within our concluding paragraph.

Thank you for consideration of our revised manuscript, which we hope you will find acceptable for publication in PLOS ONE.

Yours sincerely,

Elaine M. Leslie, Ph.D.

---

## [Editor Report · Decision Letter 1]

25 Jul 2024

Glutathione transferase P1 is modified by palmitate

PONE-D-24-21632R1

Dear Dr. Leslie,

We’re pleased to inform you that your manuscript has been judged scientifically suitable for publication and will be formally accepted for publication once it meets all outstanding technical requirements.

Kind regards,

Mohd Akbar Bhat

Academic Editor

PLOS ONE

Additional Editor Comments (optional):

The authors have amended the manuscript as per the reviewer comments.
---

## [Editor Report · Acceptance letter]

30 Jul 2024

PONE-D-24-21632R1 

PLOS ONE

Dear Dr. Leslie, 

I'm pleased to inform you that your manuscript has been deemed suitable for publication in PLOS ONE. Congratulations! Your manuscript is now being handed over to our production team.

Kind regards, 

on behalf of

Dr. Mohd Akbar Bhat 

Academic Editor

PLOS ONE